# Defective three-dimensional covalent organic frameworks for enhanced hydrogen peroxide photosynthesis and organic transformation

Tengteng Dong ®[1], Xiaohui Xu ®[2] ✉, Li Chen[1], Jiani Yang[1], Mengchao Guo[1], Mi Zhou[1], Min Xu[3], Weichao Xue ®[3], Xiancheng Ren[1], Shuang Li ®[1] ✉ & Chong Cheng ®[1,4] ✉

Covalent organic frameworks with three-dimensional networks and interconnected porous structures show promising advantages for hydrogen peroxide photocatalysis. However, 3D COFs are typically constructed from 3D-oriented knots with less conjugation and insufficient light absorption, which significantly inhibits their performance. Herein, we present a universal defect engineering approach by systematically replacing $T_d$ knots with trigonal planar ligands and modifying linear linkers with electron-withdrawing/donating groups to achieve simultaneous enhancement of light absorption and precise electronic tuning of 3D donor-acceptor structures. Experimental results and theoretical analysis reveal that the optimized 3D COF with planar ligands induced defects and fluorine functional groups (COF-300-D-F), which achieve an $H_2O_2$ production rate of 19.09 mmol g$^{-1}$ h$^{-1}$ and apparent quantum yield of 11.95% at 400 nm with benzyl alcohol as sacrificial agent. Moreover, the material maintains long-term stability during continuous operation exceeding 96 hours and exhibits high activity in photocatalytic benzylamine coupling reactions.

Hydrogen peroxide ($H_2O_2$) represents a critical industrial chemical with widespread applications ranging from biological treatment to organic synthesis[1]. Current industrial production predominantly relies on the energy-intensive anthraquinone process, which not only demands noble metal catalysts but also generates hazardous byproducts[2]. The photocatalytic route for $H_2O_2$ synthesis from water and oxygen presents an attractive and sustainable alternative, leveraging solar energy under ambient conditions while minimizing waste generation[3]. This approach combines several advantages: utilization of renewable energy inputs, operation under mild

conditions, exceptional product selectivity, and elimination of hazardous precursors[4]. Covalent organic frameworks (COFs) have emerged as particularly promising candidates for photocatalytic applications due to their unique combination of structural precision[5–7], porosity, and photochemical stability[8,9]. As crystalline, metal-free polymers, COFs offer exceptional structural tunability through modular assembly of diverse building blocks, enabling precise control over optical absorption, charge transport properties, and mass diffusion characteristics[10]. Their well-defined periodic architectures facilitate both light harvesting and charge carrier

[1]College of Polymer Science and Engineering, State Key Laboratory of Advanced Polymer Materials, Sichuan University, Chengdu, China. [2]Department of Medical Ultrasound, Frontiers Science Center for Disease-Related Molecular Network, West China Hospital, Sichuan University, Chengdu, China. [3]College of Chemistry, Sichuan University, Chengdu, China. [4]Department of Endodontics, State Key Laboratory of Oral Diseases, National Center for Stomatology, West China Hospital of Stomatology, Sichuan University, Chengdu, China. ✉e-mail: xiaohuixu@scu.edu.cn; shuang.li@scu.edu.cn; chong.cheng@scu.edu.cn

mobility, making them ideal platforms for artificial photosynthesis applications.

Current research in COF-based photocatalysis has predominantly focused on two-dimensional (2D) architectures[11–14]. These 2D COFs typically assemble through π-π stacking of planar conjugated organic linkers, which serve dual functions as efficient light-harvesting units and charge transport channels[15–17]. Significant progress has been made through strategic modifications of these systems, including band structure engineering to optimize redox potentials[18,19], donor-acceptor design to enhance intralayer charge separation[20,21], and linkage optimization to improve structural stability[22]. Such approaches have demonstrated notable success in photocatalytic $H_2O_2$ production. However, these advancements primarily address in-plane structural features while neglecting critical interlayer limitations. The inherent weak interactions between stacked layers often result in structural instability and poor interlayer charge transport, presenting fundamental challenges that remain largely unresolved in current 2D COF designs.

3D COFs feature unique characters and usually require at least one 3D-oriented building unit, such as those knots with $T_d$-symmetry, to connect with another linker[23–27]. Unlike their 2D counterparts, these 3D architectures maintain structural integrity through covalent bonding while providing multidimensional pore systems that enhance catalytic accessibility[28–30]. However, the tetrahedral or higher connectivity of these nodal units inherently limits π-conjugation, compromising their light-harvesting capabilities. Recent attempts to address this limitation through complex hexa- or octa-topic connectors or planar $C_3$ or $C_4$ symmetric linkers have yielded only modest photocatalytic improvements[31–33]. These approaches present additional challenges: the intricate synthesis of such building blocks hampers scalability, while the complicated structure impedes rational electronic tuning to optimize redox potentials. The combined effects of compromised light absorption, synthetic complexity, and limited functionalization opportunities concurrently restrict the practical application of 3D COFs in photocatalysis.

Here, we present a universal defect engineering strategy that introduces planar light-harvesting units into 3D **dia** topology COFs by partially replacing the $T_d$ knots with $C_3$ planar ligands but preserving their underlying network (Fig. 1 and Supplementary Fig. 1). By systematically replacing tetrahedral nodes with trigonal planar ligands and modifying linear linkers with electron-withdrawing/donating groups, we achieve simultaneous enhancement of light absorption and precise electronic tuning. This approach creates 3D donor-acceptor networks that promote efficient charge separation, ultimately boosting photocatalytic performance. Applied to both COF-300 and COF-320, the yield of $H_2O_2$ produced by this strategy is competitive. The structural similarity between tris(4-aminophenyl)amine (TAPA) and tetrakis(4-aminophenyl)methane (TAM) enables their partial substitution, while fluoride-functionalized terephthalaldehyde (BDA-F) linkers further optimize electronic properties. In combination with partially replacing $T_d$ knot (TAM) with TAPA and the use of BDA-F linker, the resulting COF-300-D-F demonstrates competitive performance, achieving an $H_2O_2$ production rate of 19.09 mmol g$^{-1}$ h$^{-1}$ at pH 3 with an AQY of 11.95% at 400 nm in the two-phase system of benzyl alcohol (BnOH) and deionized water. The covalently bonded 3D architecture contributes to structural and performance stability, maintaining activity over 96 hours of continuous operation. It also shows competitive performance in benzylamine coupling reactions.

## Results

### Design and synthesis of COF-300-D and COF-300-D-R
COF-300 was synthesized via solvothermal imine condensation of TAM and terephthalaldehyde (BDA) under 120 °C for 3 days to give yellowish powders (Fig. 2a and Supplementary Fig. 2). Noticed that TAPA molecules featured similar size and configurations with part of

TAM molecules, it would be geometrically possible for replacing TAM with TAPA (Fig. 1). By mixing BDA, TAM and TAPA (from 5% to 20% molar ratio versus TAM) during solvothermal reaction, a series of COFs with different contents of TAPA (COF-300-D) can be obtained, COF-300-D-5%, COF-300-D-10%, COF-300-D-15% and COF-300-D-20% (D means defects induced by TAPA molecules), respectively (Fig. 2b and Supplementary Figs. 3–4). The successful introduction of TAPA was confirmed by Fourier transform infrared spectra (FTIR, Fig. 2d and Supplementary Figs. 5–6). A new peak at 1697 cm$^{-1}$ represented the C = O vibration stretch from the unreacted formyl group in BDA that was left by the replacement of the TAPA molecule, while the peak at 1624 cm$^{-1}$ represented the C = N vibration, indicating the formation of imine linkages[34,35].

The quantitative replacement of TAPA was further verified by both $^1H$ and $^{13}C$ nuclear magnetic resonance spectroscopy (NMR) (Fig. 2e, f and Supplementary Figs. 7–22). Two new peaks at 142.41 and 132.18 ppm that were ascribed to TAPA were found in COF-300-D-15% compared to the original COF-300 in solid-state cross-polarization $^{13}C$ NMR (peaks 18 and 21 in Fig. 2e), which directly indicated the existence of TAPA. Extra peaks from TAPA at 146.28, 125.16, and 125.38 ppm in liquid $^{13}C$ NMR collected from the digested sample of activated COF-300-D-15% confirmed the same conclusion with better signal-to-noise ratio and resolution (Fig. 2f, Methods and Supplementary Figs. 7–14). Liquid $^1H$ NMR was used to quantify the content of TAPA in the COF-300-D series, and it was found that the actual amount of TAPA in the materials was linear with the input ratio (Supplementary Figs. 15–23). Nevertheless, to introduce functionalized BDA linkers (2,5-difluoro-1,4-benzenedicarboxaldehyde, BDA-F; 2,1,3-benzothiadiazole-4,7-dicarboxaldehyde, BDA-Tz; 2,5-dimethoxy-1,4-benzenedicarboxaldehyde, BDA-OMe; and 2, 5- dihydroxy-1, 4- benzenedicarboxaldehyde, BDA-OH) into COF-300-D with TAPA amounts fixed at 15%, a similar multivariate strategy was employed by mixing BDA with different linkers to obtain the electronically tuned COF-300-D, named COF-300-D-R (R, -F, -Tz, -OMe, or -OH) (Fig. 2c and Supplementary Figs. 24–25). Both FTIR and NMR were used to confirm the successful synthesis of COF-300-D-R (Fig. 2d, e and Supplementary Figs. 12–14).

The crystallinity and the underlying topology of the parent COF were well preserved after this multivariate ligand doping to form the COF-300-D, COF-300-D-R, COF-320-D-15%, and COF-320-D-F series (Fig. 2g and Supplementary Figs. 26–31). Sharp peaks were clearly observed in the powder X-ray diffraction (PXRD) patterns at low doping ratios, indicating high crystallinity, and no extra peaks from other phases were found (Fig. 2g)[36,37]. Once the doping amount reaches 25% or more, it may lead to the loss of long-range ordered structure. (Supplementary Fig. 26). 2D SAXS images provided accurate diffraction intensities in the low q-range due to their transmission optics geometry (Supplementary Figs. 32–33). Pawley refinements were then conducted under the same tetragonal $I4_1/a$ space group against these experimental SAXS patterns, yielding Rwp and Rp values for all COFs of less than 3%. COF-300-D with different TAPA amounts (5%, 10%, 15%, 20%) displayed systematic lattice parameter changes compared to COF-300, from 27.24 to 27.10 Å in $a$ and from 7.51 to 7.56 Å in $c$ (Supplementary Fig. 33 and Supplementary Table 1). These variations can be attributed to the defects introduced by TAPA molecules. For COF-300-D-F, the lattice parameters changed slightly from 27.12 to 27.08 Å in a and from 7.56 to 7.57 Å in $c$, compared to COF-300-D-15%. The full width at half maximum (FWHM) values reflected the coherence of the structure across the entire crystal sample. It is worth noting that with the increase in TAPA content, the FWHM increases gradually (Supplementary Fig. 34), indicating that structural disorder accompanies the increment, in which the introduction of functional nodes and crystal integrity are balanced. The relatively narrow FWHM of 200, 220, and 400 reflections at low $q$ range further confirmed the preservation of the integrity of the COF local structure as TAPA units were introduced.

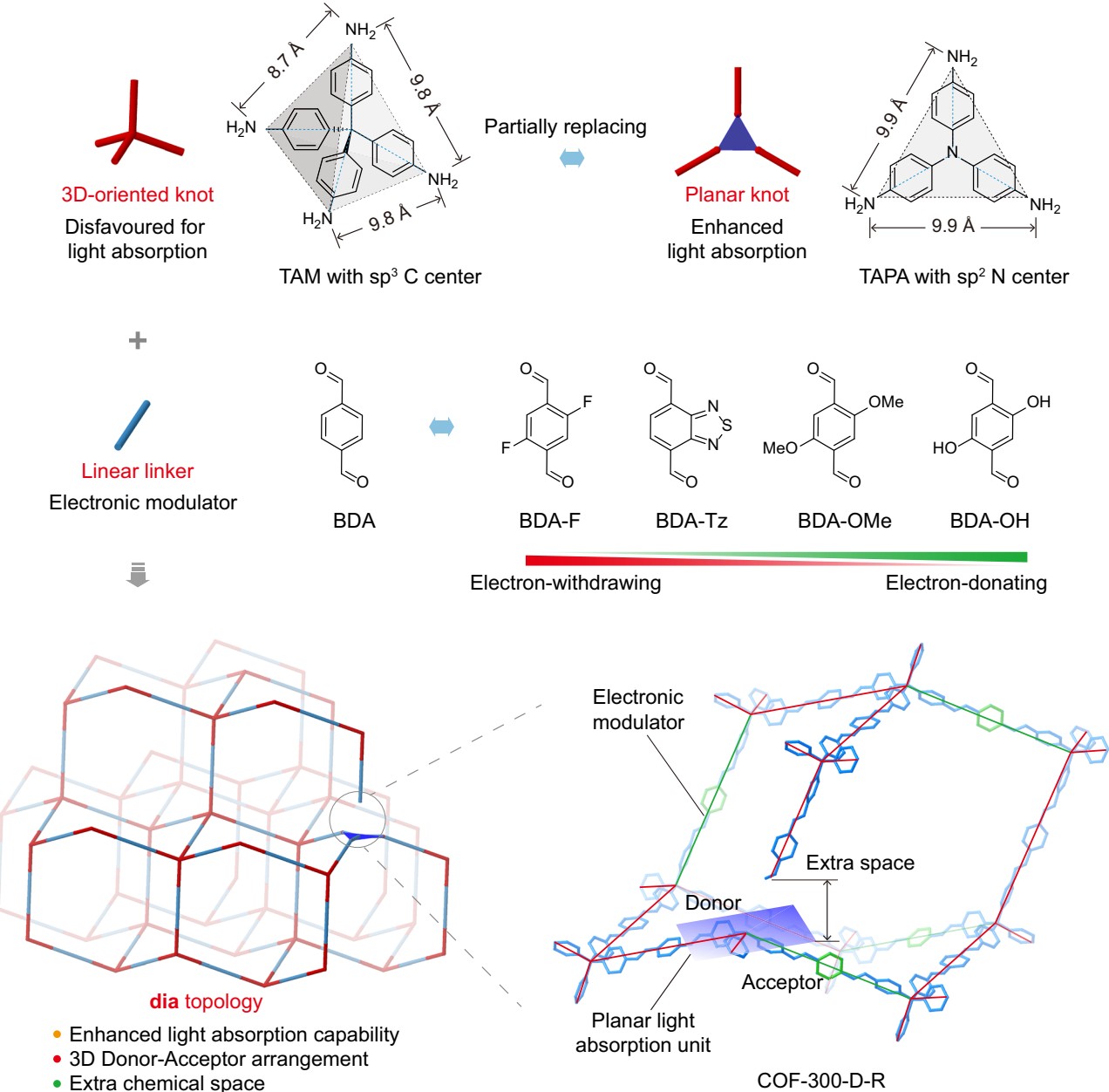

**Fig. 1 | Illustration of the molecular engineering of defective 3D COFs.** Collectively replacing tetrahedral knots (TAM) with trigonal planar ligands (TAPA) and modifying linear linkers (BDA) with electron-withdrawing/donating groups (BDA-F, BDA-Tz, BDA-OMe, or BDA-OH).

Scanning electron microscopy (SEM) revealed that all COF-300-D and COF-300-D-R series were spindle-like single crystals with negligible morphology alteration compared to the parent COF-300 (Supplementary Figs. 35–36). The atomic distribution of TAPA in COF-300-D was hard to determine using X-ray or electron diffraction techniques due to both occupational and displacement disorders. Considering the fluorescence nature of TAPA linkers, we applied laser scanning confocal microscopy (LSCM) to visualize the distribution of TAPA linkers throughout the single crystal of COF-300-D-15%. The precise distribution of TAPA across COF-300-D-15% single-crystal was mapped layer by layer at different focal depths with about 0.2 μm intervals to reconstruct 3D tomography (Fig. 2h). Red fluorescence was observed in all layers under excitation light of 405 nm, indicating the homogeneous and random distribution of TAPA across the whole crystal, while COF-300 exhibits no fluorescence under the same condition (Supplementary Figs. 37–38). Additionally, the elemental dispersion spectroscopy (EDS) mapping confirmed the

uniform dispersing of fluorine across the entire COF-300-D-F crystal, reflecting that the BDA-F linker was also homogenously distributed (Supplementary Fig. 39)[38,39].

Nitrogen isotherms were used to verify the permanent porosity of these COFs at 77 K[40]. The isotherms types of COF-300, COF-300-D-15%, and COF-300-D-F were distinctive for their unique dynamic responsive structures[41], and their Brunauer-Emmett-Teller (BET) surface areas were calculated to be 595, 1288, 1463 $m^2\,g^{-1}$, while the pore volume was 0.67 $cm^3\,g^{-1}$, 0.75 $cm^3\,g^{-1}$, and 0.81 $cm^3\,g^{-1}$ at $P/P_O = 0.95$, respectively (Fig. 2i and Supplementary Fig. 40). The analysis of pore size distribution by the nonlocal density functional theory model shows that the average pore size of COF-300, COF-300-D-15%, and COF-300-D-F is about 1.08 nm (Supplementary Fig. 41). The increments of pore volume in COF-300-D-15% and COF-300-D-F were likely induced by the extra space created by the partial replacement of TAM with TAPA, which was more conducive for mass transfer. Notably, the introduction of additional void spaces through the defect engineering did not

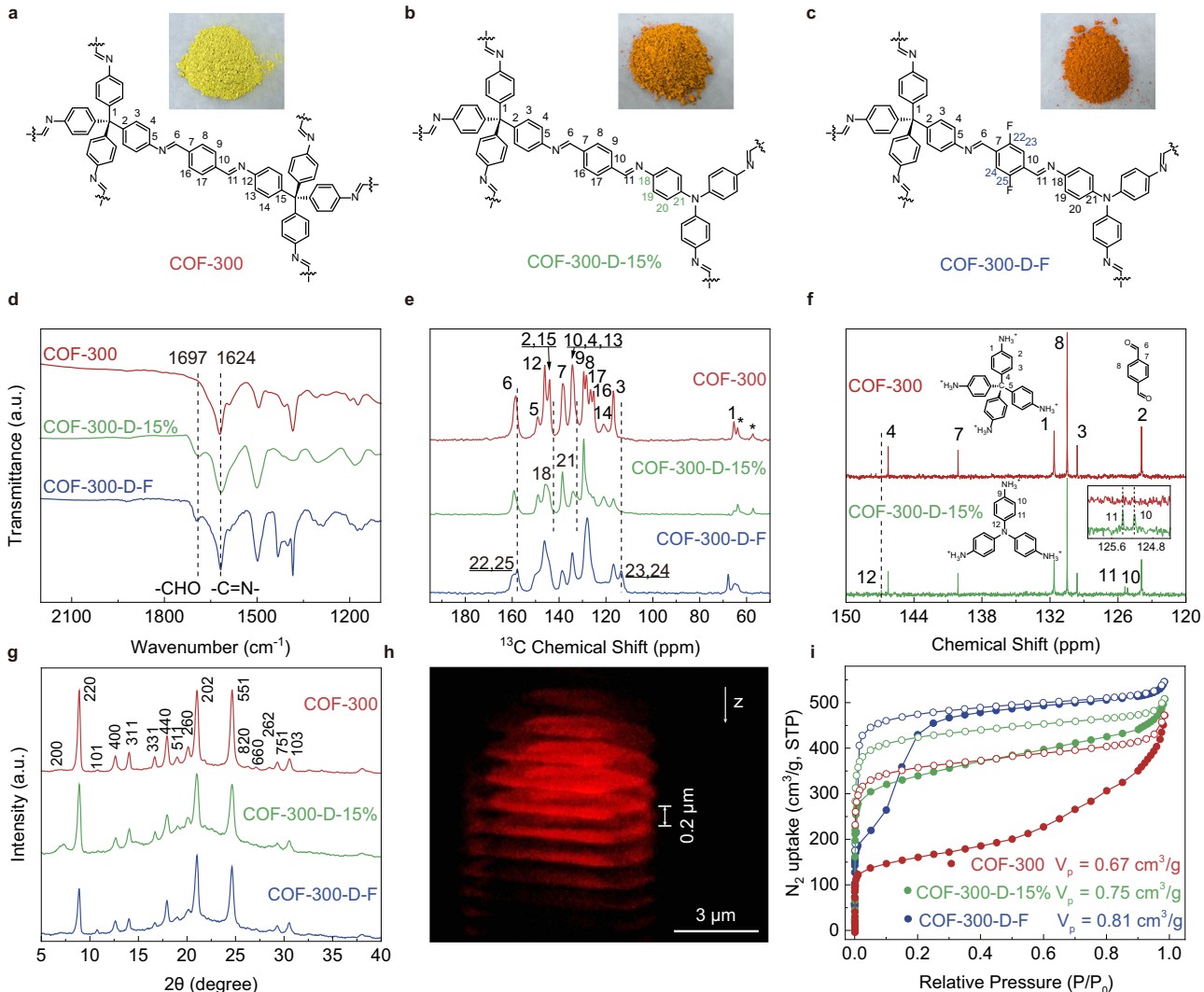

**Fig. 2 | Structural characterization of functional 3D COFs.** Representative structural unit of the parent COF (COF-300) (**a**), the COF with TAPA content of 15% (COF-300-D-15%) (**b**), and the optimized 3D COF with planar ligands induced defects and fluorine functional groups (COF-300-D-F) (**c**). Insets: corresponding optical photograph. **d** FT-IR spectra of COF-300, COF-300-D-15%, and COF-300-D-F. **e** Solid-state $^{13}$C CP/MAS NMR spectrum of COF-300, COF-300-D-15%, and COF-300-D-F and corresponding carbon signal assignment (*, sideband). **f** Solution-state $^{13}$C NMR spectroscopy of acid-digested COF-300 and COF-300-D-15%. **g** PXRD patterns of COF-300, COF-300-D-15%, and COF-300-D-F were experimentally obtained. **h** 3D tomography of COF-300-D-15% by laser scanning confocal microscopy. Consecutive LSCM sections of COF-300-D-15%. **i** Nitrogen sorption isotherms of COF-300, COF-300-D-15%, and COF-300-D-F (solid circle, adsorption; open circle, desorption). Measurement (**h**) was repeated at least three times independently with similar results. a.u. indicates the arbitrary units in (**d**, **g**). Source data are provided as a Source data file.

compromise the thermal stability of COF frameworks, with treated materials retaining decomposition temperatures comparable to pristine samples under inert conditions (Supplementary Figs. 42–43)[42,43].

## Energy levels and charge separation analysis

Ultraviolet-visible diffuse reflectance spectroscopy (UV-vis DRS) was employed to characterize the light-harvesting properties of these materials, evaluating the effectiveness of the defect engineering strategy[44]. Compared to COF-300, the COF-300-D series showed huge redshifts of light absorption while the TAPA content increased (Fig. 3a and Supplementary Fig. 44). COF-300-D-20% displayed the largest improvement of light absorption maximum from 483 nm to 621 nm, which indicated that the introduction of planar TAPA molecules enhanced light absorption effectively. Furthermore, UV-vis absorption spectroscopy demonstrated a substantial redshift in COF-320-D-15% compared to pristine COF-320 (Supplementary Fig. 45), confirming the efficacy of the unique structure obtained

from defect engineering and its potential for broad implementation in 3D COF functionalization.

The optical band gaps also reduced drastically from 2.67 eV to 2.21, 2.19, 2.20, and 2.14 eV for COF-300-D-5%, -10%, -15%, and -20%, respectively, as derived from Tauc plots by using the Kubelka-Munk equation from DRS analysis (Fig. 3b and Supplementary Figs. 44 and 46–48). Among the three COFs, COF-300-D-F exhibits the most negative conduction band (CB) level (−0.55 V vs. normal hydrogen electrode (NHE)), surpassing COF-300-D-15% (−0.52 V) and COF-300 (−0.44 V). This trend suggests that the electron-acceptor sites in COF-300-D-F possess the strongest photo-reduction capability, facilitating the oxygen reduction reaction (ORR) to $H_2O_2$ via the $\cdot O_2^-$ intermediate (−0.33 V) (Fig. 3b). In addition, the valence band levels (VB) of COF-300-D-F, COF-300-D-15%, and COF-300 are +1.61, +1.68, and +2.23 V (versus NHE), respectively, which are positive enough to allow the oxidation reaction of benzyl alcohol (BnOH) to benzaldehyde (BD) (Fig. 3b)[45,46]. The partial density of states (PDOS) also shows the

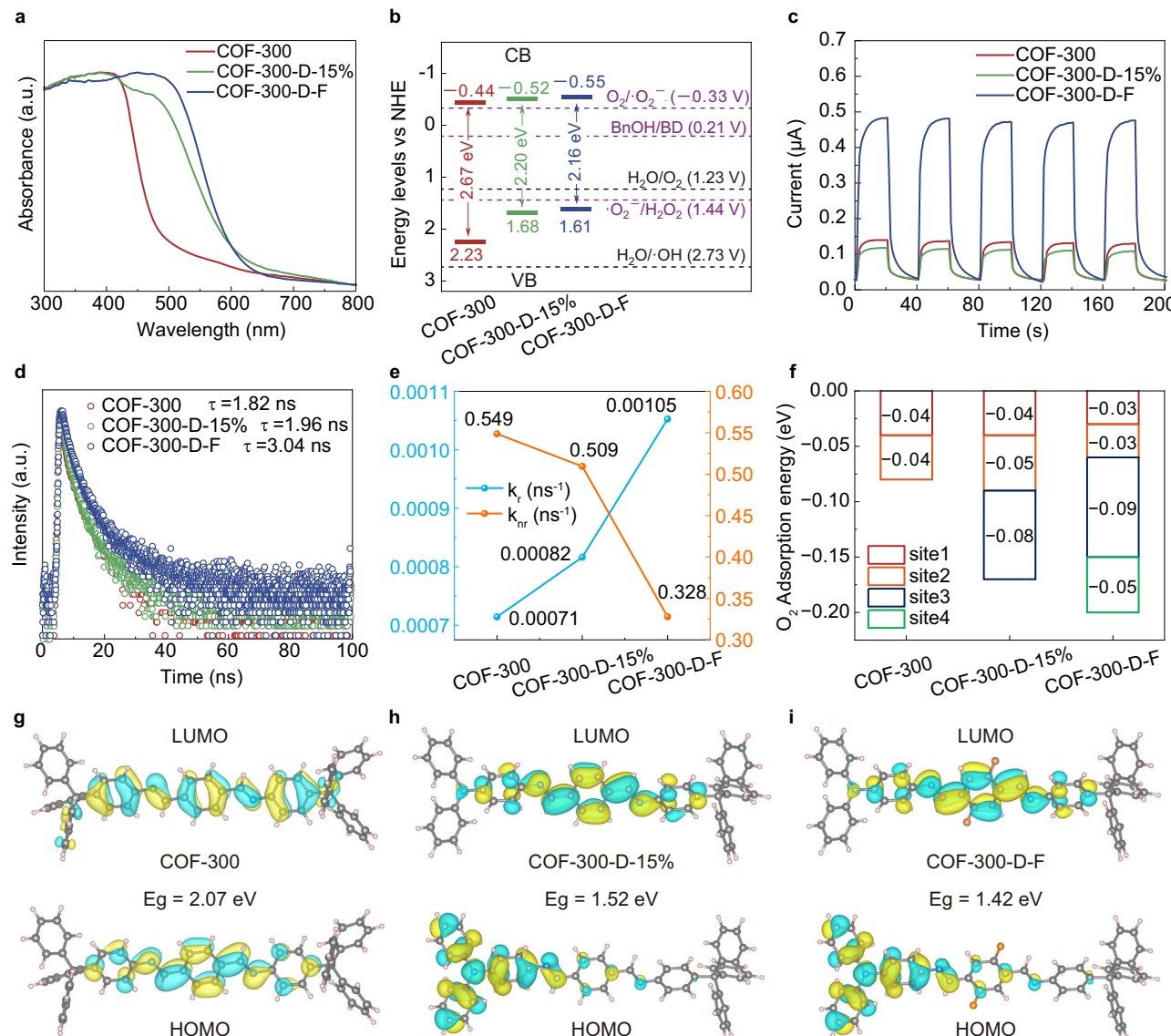

**Fig. 3 | Photophysical and electrochemical measurements. a** UV-vis DRS of COF-300, COF-300-D-15%, and COF-300-D-F. **b** Energy band values of COF-300, COF-300-D-15%, and COF-300-D-F. **c** Transient photocurrents of COFs under $\lambda > 420$ nm. **d** Fluorescence lifetime decay spectra of COF-300 ($\tau_{ave1}$), COF-300-D-15% ($\tau_{ave2}$), and COF-300-D-F ($\tau_{ave3}$) at the excitation wavelength of 480 nm and their respective maximum emission wavelengths (600 nm, 650 nm, and 650 nm, respectively).

**e** The carrier radiative recombination rate ($k_r$) and nonradiative recombination rate ($k_{nr}$) obtained by calculation. **f** Adsorption energies of $O_2$ on three COFs. Distributions of LUMO/HOMO for the structural segments of COF-300 (**g**), COF-300-D-15% (**h**), and COF-300-D-F (**i**). F, N, C, and H atoms are marked in orange, blue, grey and pink, respectively. a.u. indicates the arbitrary units in (**a**, **d**). Source data are provided as a Source data file.

same trend (Supplementary Fig. 49). By virtue of the electronic configuration modulator, COF-300-D-F exhibited an enhanced photocurrent density (wavelength $\lambda > 420$ nm) (Fig. 3c and Supplementary Fig. 50), indicating the high efficiency of photoinduced charge carrier generation[47].

In the steady-state photoluminescence spectrum collected under the excitation wavelength of 480 nm (PL, Supplementary Figs. 51–52), it can be observed that, compared to COF-300, the fluorescence intensity gradually increases for COF-300-D-F[48]. We further measured its photoluminescence quantum yield (PLQY) and fluorescence lifetime. As shown in Supplementary Fig. 53, the PLQY of COF-300, COF-300-D-15%, and COF-300-D-F are 0.13%, 0.16%, and 0.32%, respectively. COF-300-D-F has a significantly longer lifetime of the charge-separated state ($\tau_{ave3} = 3.04$ ns) than COF-300 ($\tau_{ave1} = 1.82$ ns) and COF-300-D-15% ($\tau_{ave2} = 1.96$ ns) (Fig. 3d), implying more efficient charge separation. Such an excited state lifetime reflected the overall carrier

recombination rate. Through the quantitative analysis of fluorescence kinetic parameters, the carrier nonradiative recombination rate ($k_{nr}$) and radiative recombination rate ($k_r$) are further calculated according to the formulas $\text{PLQY} = \frac{k_r}{k_r + k_{nr}}$ and $\tau_{ave} = \frac{1}{k_r + k_{nr}}$ (Fig. 3e)[48,49]. All samples are excited by a 480 nm laser. The decrease in nonradiative recombination and the increase in radiative recombination indicate that, compared with COF-300, the nonradiative recombination of COF-300-D-F is inhibited, resulting in higher visible light activity. This is due to the introduction of the light absorption unit (TAPA) and D-A structure in COF-300-D-F, which can effectively improve the overall photocatalytic efficiency.

Density functional theory (DFT) calculations were performed to investigate the interactions between reaction substrates and the electronic structures of these COFs (Supplementary data 1). With the introduction of TAPA and BDA-F units, the number of oxygen adsorption sites increases, enhancing the oxygen affinity and

facilitating the activation of oxygen for subsequent transformations (Fig. 3f and Supplementary Fig. 54)[50]. Moreover, the extra channels created through the partial substitution of TAM with TAPA facilitate $O_2$ transport during the reaction. To reveal the frontier molecular orbitals, we analyzed the highest occupied molecular orbital (HOMO) and the lowest unoccupied molecular orbital (LUMO) in particular. According to the results, the HOMO and LUMO orbitals of COF-300-D-F are localized on the electron-donor TAPA unit and the electron-acceptor BDA-F linker, respectively, while the HOMOs and LUMOs are rather evenly distributed in COF-300 (Fig. 3g–i)[51]. This meaningful spatial separation can effectively inhibit electron-hole recombination, extend electron/hole lifetime, and promote photocatalytic reactions. Electrostatic potential (ESP) distribution calculations confirm the same donor-acceptor distribution (Supplementary Fig. 55). To quantify the charge transfer between donor and acceptor units of different COFs, Bader charge calculation was performed and shown in Supplementary Fig. 56. They indicate that the BDA-F linker in COF-300-D-F exhibits an increased charge density compared to the BDA linker in COF-300 and COF-300-D-15%, with the charge transfer number increasing from 0.9 | e| to 1.1 |e | , significantly enhancing the intramolecular polarity. These results indicate that a 3D donor-acceptor structure between different functional units is successfully constructed, and the ORR occurs at the BDA-F units. It is worth noting that, given the inherent challenges in computationally modelling the exact dynamic structure of the COF under operational photocatalytic conditions, these calculated results should be considered as qualitative trends rather than definitive quantitative predictions.

## Photocatalytic $H_2O_2$ production

We initially selected benzyl alcohol (BnOH) as a hole sacrificial agent to evaluate the intrinsic activity of the COFs in the photocatalytic production of $H_2O_2$ under visible light ($\lambda > 420$ nm). The $H_2O_2$, as the $O_2$ reduction product, is dissolved in water, while the BnOH, as a hole scavenger, is oxidized into value-added benzaldehyde. The photocatalytic $H_2O_2$ production rates of the COF-300-D series increased first and then decreased with the increase of TAPA in the range of 0% ~ 20%, among which COF-300-D-15% showed the highest rate of 1.74 mmol $g^{-1}$ $h^{-1}$ (Fig. 4a and Supplementary Fig. 57). The observed trend may result from a competition between enhanced photon harvesting and a progressive amorphization of the COF scaffold induced by the incorporation of defective engineering (Supplementary Fig. 26)[52–56]. Additionally, we employed the same synthetic protocol to substitute 15% of TAM knot in COF-300 with tri(4-aminophenyl)methane (MTA), thereby achieving a 3D structure, COF-300-MTA, that concurrently introduces defects but lack of planar light absorption units (Supplementary Fig. 58). COF-300-MTA displayed a photocatalytic $H_2O_2$ production rate of 1.18 mmol $g^{-1}h^{-1}$, which was less efficient than that of COF-300-D-15%, indicating the critical role of planar structure for photocatalytic activity (Supplementary Fig. 59).

Moreover, the $H_2O_2$ production rates were significantly influenced by the introduction of functionalized BDA linkers bearing electron-withdrawing or donating groups to form COF-300-D-R, thereby systematically modulating the electronic configuration. After 180 min of light irradiation, it showed a production rate trend with -F > -Tz > -OMe > -OH, indicating the strong electron-withdrawing linkers were important for the formation of 3D donor-acceptor structures (Fig. 4b). COF-300-D-F exhibited a photocatalytic rate of 10.09 mmol $g^{-1}$ $h^{-1}$, which was 21.6 and 5.8 times that of COF-300 (0.47 mmol $g^{-1}$ $h^{-1}$) and COF-300-D-15% (1.74 mmol $g^{-1}h^{-1}$), respectively[33,57,58]. In control experiments to verify whether the introduction of TAPA linkers in COF-300-D-F is necessary, we synthesized the BDA-F-based crystalline COF without TAPA linkers. We named it COF-300-F (Supplementary Fig. 60). The results showed that COF-300-F exhibited a much lower photocatalytic activity (1.98 mmol $g^{-1}$ $h^{-1}$), which was less than 20% of that of COF-300-D-F (Fig. 4b), indicating

that the introduction of TAPA ligand was critical. The synergy between light-absorbing units and electron modulation units is efficient for photocatalysis.

We also compared its $H_2O_2$ production efficiency in pure water and with different hole sacrificial agents (Fig. 4c). It is shown that the yield of $H_2O_2$ for COF-300-D-F is 1.20 mmol $g^{-1}$ $h^{-1}$ in pure water without any organic sacrificial agent. Although the efficiency is lower than that of the system using an organic sacrificial agent, it provides the possibility for its application in a more practical scenario. Photocatalytic experiments of COF-300-D-F under different sacrificial agents showed that BnOH is the best, which indicates that a two-phase reaction system composed of water and BnOH could effectively facilitate the reaction, because the active center of COFs remains in the BnOH phase, and the generated $H_2O_2$ quickly diffuses into the water. Furthermore, COF-320-D-F exhibited a rate of 4.24 mmol $g^{-1}$ $h^{-1}$ for photocatalytic $H_2O_2$ production, which was 7.6 and 2.6 times that of COF-320 (0.56 mmol $g^{-1}$ $h^{-1}$) and COF-320-D-15% (1.63 mmol $g^{-1}$ $h^{-1}$), respectively (Fig. 4d). This confirms the potential of the defect engineering strategy in the functionalization of 3D COF.

In addition to its competitive $H_2O_2$ generation activity, COF-300-D-F also showed good photocatalytic recyclability; no obvious drops in the production rate were observed after 5 consecutive cycles (Fig. 4e). The stability of COF-300-D-F was rigorously assessed after five cycles of photocatalytic $H_2O_2$ production using XRD, FT-IR, and SEM techniques (Supplementary Figs. 61–64). The results demonstrated the retention of crystallographic features and morphological characteristics, confirming the robustness of the framework under repeated use. This indicates the competitive chemical stability characteristic of 3D bonded covalent structures, as opposed to the layered stacking of 2D materials. COF-300-D-F also showed stable performance across a wide pH range from 3 to 11. It's worth noting that the production rate reached 19.09 mmol $g^{-1}$ $h^{-1}$ at pH = 3, which might be boosted by the acidified imine linkages (Fig. 4f). With its structural stability and an AQY of 11.9% at 400 nm, COF-300-D-F demonstrates performance that is competitive among reported 3D COFs and 2D COFs constructed with similar linkers (Fig. 4g and Supplementary Table 2).

It is worth noting that this synthesis strategy endows COF-300-D-F with structural and performance stability during photocatalysis, which can run continuously for more than 96 h without any noticeable decline. The final concentration of $H_2O_2$ reaches 0.10 wt%. It can be used for daily applications, such as oral cleaning directly (Fig. 4h and Supplementary Fig. 65). It is important to note that a slight increase in the apparent $H_2O_2$ production rate is observed after 24 h of continuous photocatalysis (Fig. 4h). This behavior can be attributed to the progressive pore wetting and enhanced mass transport within the 3D COF framework, together with sustained stirring that promotes a dynamic emulsified interfacial state, thereby facilitating interfacial oxygen reduction and $H_2O_2$ formation (Supplementary Fig. 66)[6,59–61]. The characterization of PXRD, FTIR, and SEM after 96 h of photocatalytic operation showed that the structural integrity and morphological stability of COF-300-D-F were maintained after long-term photocatalytic production of $H_2O_2$. No obvious degradation was observed compared with that before photocatalytic reaction (Supplementary Fig. 67). However, after 120 h of photocatalytic reaction, the long-range ordered structure of COF-300-D-F underwent obvious degradation. This was evidenced by the disappearance of sharp peaks in the PXRD pattern and the morphological deterioration of the particles observed by SEM. In contrast, the IR spectrum showed that C=N linkages (1624 cm$^{-1}$) were still retained, collectively suggesting that the crystalline framework was largely collapsed into short-range domains after photocatalysis for 120 h. The solution extracted after the 96 h reaction was further analyzed by gas chromatography (GC), $^1$H NMR, and $^{13}$C NMR (Supplementary Figs. 68–70). The results showed that BnOH was selectively oxidized to benzaldehyde, while benzoic acid was not detected.

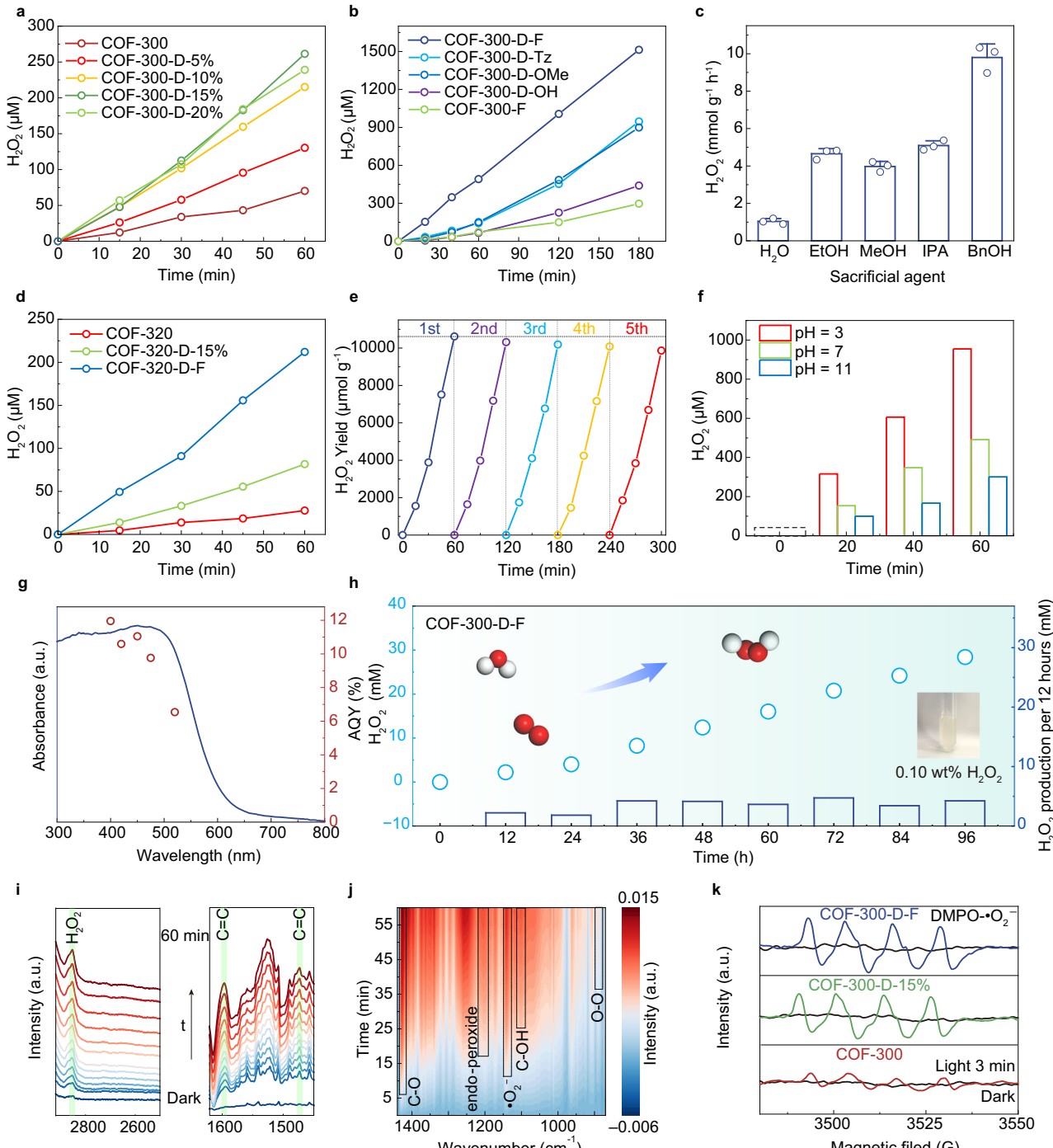

**Fig. 4 | H₂O₂ evolution performance of functional 3D COFs. a** Photocatalytic activity of COF-300 and COF-300-D-5%, -10%, -15% and -20% for H₂O₂ generation (3 mg of COFs in 20 mL of water/BnOH (9:1) at 25 °C and $\lambda > 420$ nm). **b** Photocatalytic activity of different functional side chain groups of COF-300-D-R and COF-300-F without TAPA unit for H₂O₂ generation (1 mg of COFs in 20 mL of water/BnOH (9:1) at 25 °C and $\lambda > 420$ nm). All reactions were performed under these conditions unless otherwise stated. **c** H₂O₂ generation rates of the photocatalytic reaction of COF-300-D-F with pure water and different sacrificial agents (ethanol, EtOH; methanol, MeOH; isopropanol, IPA; benzyl alcohol, BnOH). **d** Photocatalytic activity of COF-320, COF-320-D-15%, and COF-320-D-F for H₂O₂ generation. **e** The H₂O₂ production yield achieved by COF-300-D-F across five reuse cycles. **f** The effects of initial solution pH on

H₂O₂ yield by COF-300-D-F under $\lambda > 420$ nm. **g** Apparent quantum yield (AQY) of COF-300-D-F at different wavelengths. (20 mg of COFs in 20 mL of water/BnOH (9:1) at 25 °C). **h** Continuous and stable production of H₂O₂ over 96 h of COF-300-D-F (2 mg of COFs in 80 mL of water/BnOH (9:1) at 20 °C and $\lambda > 420$ nm). **i** In situ DRIFTS study was performed on COF-300-D-F in an O₂/steam atmosphere, from 2920 to 2500 cm⁻¹ and 1620–1450 cm⁻¹. **j** The contour plot of operando infrared spectroscopy for COF-300-D-F. **k** 5,5-Dimethyl-1-pyrroline N-oxide (DMPO) spin trapping EPR spectra of COF-300, COF-300-D-15%, and COF-300-D-F for measuring •O₂⁻ under dark and visible light (3 min). Experiments (**a–k**) were repeated at least three times independently with similar results. a.u. indicates the arbitrary units in (**g**, **i–k**). Source data are provided as a Source data file.

To reveal the photocatalytic mechanisms of $H_2O_2$ production by COF-300-D-F under visible light irradiation, radical capture experiments were performed. A significant decrease in $H_2O_2$ yield is observed with Ar purging ($O_2$ removal) or the addition of $AgNO_3$ (electron scavenger) into the reaction system, while the addition of tert-butanol (TBA), a hydroxyl radical (•OH) trapping agent, showed limited influence on $H_2O_2$ production (Supplementary Figs. 71–73). These results indicate that the $H_2O_2$ synthesis in this reaction mainly involves the $O_2$ reduction to produce $•O_2^-$ and its consequent transformation, which is a two-step one-electron indirect pathway: $O_2 + e^- \rightarrow •O_2^-$, $•O_2^- + 2H^+ + e^- \rightarrow H_2O_2$.

To confirm the generation of reactive species and clarify the reaction mechanism, in situ diffuse reflectance infrared Fourier transform spectroscopy (DRIFTS) analysis was performed on COF-300-D-F in an $O_2$/steam atmosphere (Fig. 4i, j and Supplementary Fig. 74). Notably, the O–H bending vibration of $H_2O_2$ ($2851\,cm^{-1}$) showed time-dependent enhancement during photoirradiation, indicating $H_2O_2$ production (Fig. 4i)[62]. The intensity of characteristic peaks for endo-peroxide ($1204\,cm^{-1}$), $•O_2^-$ ($1141\,cm^{-1}$), and $O–O$ ($889\,cm^{-1}$) progressively increased, confirming $O_2$ adsorption and subsequent two-step reduction to form superoxide radicals[63]. Combined with the support of electron paramagnetic resonance (EPR), the indirect oxygen reduction reaction pathway can be confirmed by detecting $•O_2^-$ (Fig. 4k and Supplementary Figs. 75–76)[64,65]. Upon 3 min visible light irradiation, COF-300-D-15% and COF-300-D-F exhibited stronger characteristic signals indicative of $•O_2^-$ than COF-300. In contrast, no obvious signals were detected in the dark, confirming the formation of $•O_2^-$ during the reaction. This indicates that both COF-300-D-15% and COF-300-D-F exhibit higher photogenic charge-separation efficiency and oxygen-activation ability.

## Photocatalytic benzylamine oxidative coupling activity

To explore the broader applicability of the proposed defect engineering strategy, the aerobic oxidation of benzylamine, a key reaction in fine chemistry, was examined (Fig. 5). The model reaction was conducted in $CH_3CN$ using benzylamine under $O_2$ at room temperature. COF-300-D-F gives a conversion rate of 98.38% in 11 h, while COF-300 and COF-300-D-15% are much less efficient (Fig. 5a). Moreover, COF-300-D-F demonstrates competitive catalytic activities compared to COF-300-D-15% and COF-300 in the oxidation of a series of benzylamine derivatives (Fig. 5a and Supplementary Figs. 77–81). We further used XRD, FT-IR, and SEM to verify the stability of the COF materials after the catalytic benzylamine coupling reaction (Fig. 5b, c and Supplementary Figs. 82–84). Post-catalytic characterization revealed that COF-300-D-F retained its structural integrity and morphological stability following photocatalytic benzylamine coupling, with no significant degradation observed compared to the COF-300. As shown in the recycling test, the COF-300-D-F exhibited competitive catalytic performance and stability across five cycles (Fig. 5d).

In addition, the time-dependent changes of substrates and products during the reaction were analyzed by in situ DRIFTS spectroscopy (Fig. 5e). It is worth noting that meaningful signal peaks of $•O_2^-$ ($1139\,cm^{-1}$) and OOH* radicals ($1368\,cm^{-1}$) were observed and obviously increased over time, shows that $•O_2^-$ and OOH* participate in the benzylamine coupling reaction. In addition, clear absorption peaks of the C=N bond ($1624\,cm^{-1}$) and the C=O bond ($1697\,cm^{-1}$) were observed, attributed to the key intermediates and products (benzylideneamine (Ph−CH=NH), benzaldehyde (Ph−CHO), and N-benzylbenzaldimine) formed during the benzylamine dehydrogenation coupling process. Furthermore, the key active species in photocatalysis were revealed through quenching experiments (Fig. 5f). The marked inhibition observed with $^1O_2$ (2,2,6,6-tetramethylpiperidine) and $•O_2^-$ (p-benzoquinone) scavengers suggests that these oxygen species are central to the reaction pathway. Depletion of electrons ($AgNO_3$) or holes (KI) similarly impaired catalysis, confirming the necessity of

photogenerated charge separation. By contrast, tert-butanol (•OH scavenger) had minimal impact, suggesting that hydroxyl radicals were not major contributors. Collectively, these findings demonstrate that the photocatalytic mechanism is governed by $^1O_2$ and $•O_2^-$, operating in concert with charge-carrier dynamics.

In summary, the potential mechanism of COF-300-D-F in the benzylamine coupling system is shown in Fig. 5g. The COF-300-D-F absorbs photons and produces electron−hole pairs at visible light, then $O_2$ reacts with photogenerated electrons to form $•O_2^-$, and $^1O_2$ is formed via energy transfer from COFs. Then, $^1O_2$ would react with benzylamine to release •OOH and form Ph−CH = NH. Concurrently, the photogenic holes oxidize the adsorbed benzylamine to form $PhCH_2NH_2^{•+}$, which deprotonates gradually by the $•O_2^-$ to convert into the more reactive Ph−CH=NH. Subsequently, Ph−CH=NH reacts with another benzylamine molecule to form benzalaniline as the final product, accompanied by the release of ammonia. The above result demonstrates that the COF-300-D-F preparation, achieved through defect engineering, exhibits a wide range of working capabilities in photocatalysis.

## Discussion

We introduce a generalizable defect engineering strategy to construct photoactive 3D COFs, where partial substitution of 3D-oriented knots with planar building blocks simultaneously creates extended π-conjugated systems and spatially defined donor-acceptor architectures. This approach addresses three critical challenges in COF-based photocatalysis: light harvesting efficiency, charge separation, and structural stability. The optimized COF-300-D-F exhibits competitive performance for photocatalytic $H_2O_2$ production, achieving a rate of $19.09\,mmol\,g^{-1}\,h^{-1}$ in a benzyl alcohol/water system. This process maintains structural and operational stability for over 96 h. The methodology's versatility is further evidenced by its efficacy in photocatalytic benzylamine coupling, suggesting broad applicability in organic transformations. This strategy offers a design approach for functional 3D COFs, with potential implications extending beyond photocatalysis to applications in energy conversion systems and selective molecular transformations. The ability to precisely control electronic and spatial properties through defect engineering provides a way to tailor porous organic materials for specific functions. However, defect introduction exhibits a critical threshold beyond which accumulated lattice strain compromises crystallinity. This strategy can be extended to other 3D COF platforms, where identifying similar compositional limits will be essential. Moving forward, designing frameworks with higher intrinsic tolerance for structural diversity or employing geometrically compatible linkers offers a promising path to multifunctional COFs that maintain structural integrity while integrating advanced functionalities.

## Methods
### Materials and reagents

Tetrakis(4-aminophenyl)methane (TAM, 98%+), tris(4-aminophenyl)amine (TAPA, 98%+), 2,5-difluoro-1,4-benzenedicarboxaldehyde (BDA-F, 95%), 2,1,3-benzothiadiazole-4,7-dicarboxaldehyde (BDA-Tz, 95%), 2,5-dimethoxy-1,4-benzenedicarboxaldehyde (BDA-OMe, 98%+), and 2,5-dihydroxy-1,4-benzenedicarboxaldehyde (BDA-OH, 98%+) were purchased from Jilin Chinese Academy of Science-Yanshen Technology Co., Ltd. Terephthalaldehyde (BDA, ≥98%), 1,4-dioxane (≥99%, AR), aniline (≥99.5%, AR), N, N-dimethylacetamide (DMA, ≥99%, AR), 1,2-dichlorobenzene (DCB, ≥99%), pyridine (≥99.5%, AR) and Acetic acid (≥99%) were purchased from Shanghai Aladdin Co., Ltd. Deuterium chloride (DCl, 38 wt% in $D_2O$) and dimethyl sulfoxide-$d_6$ (DMSO-$d_6$, 99.9%) were obtained from Energy Chemical. Tetrahydrofuran (THF, AR) and methanol (AR) were sourced from Chengdu Jixin Technology Co., Ltd. All chemicals were used as received without further purification.

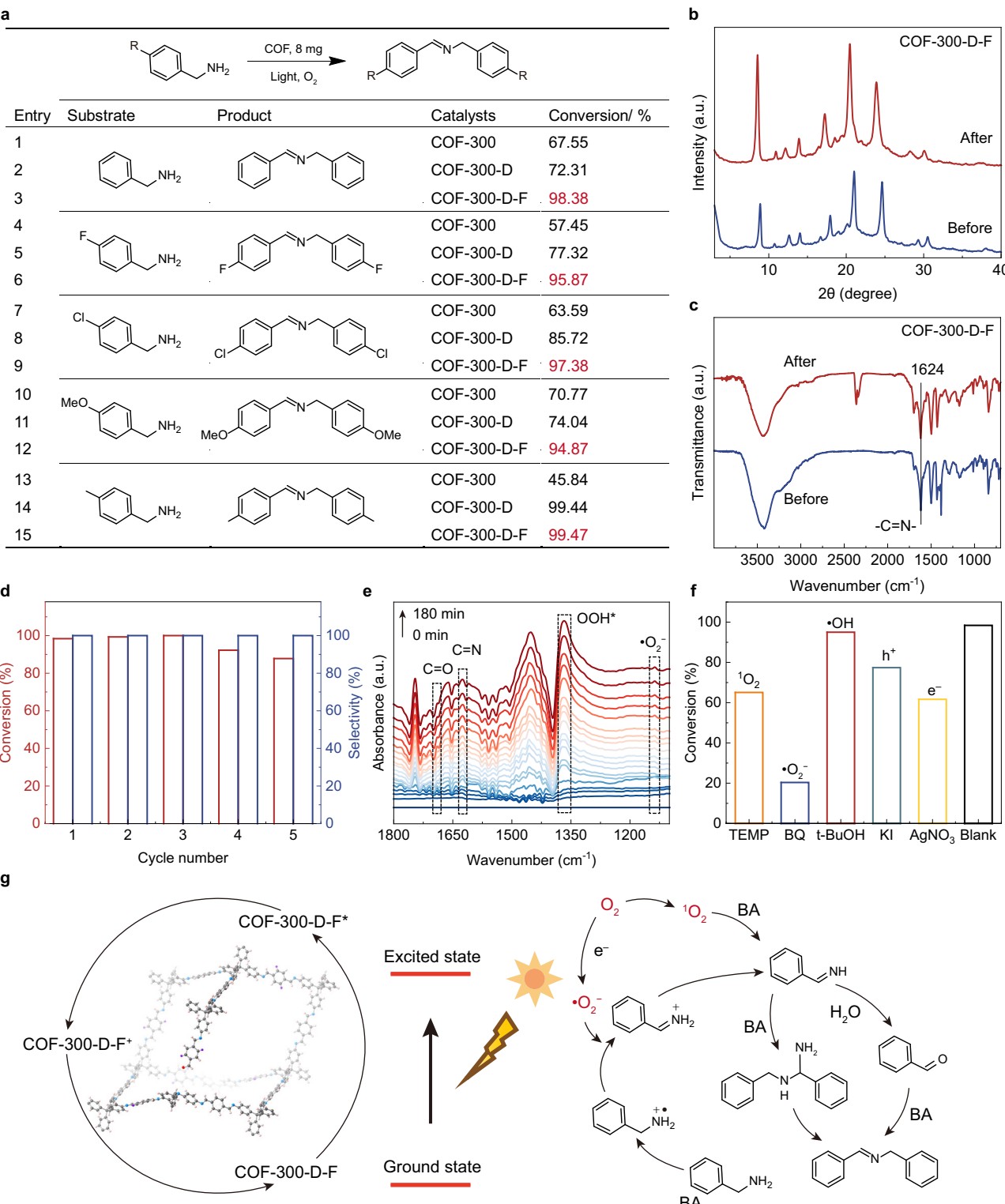

**Fig. 5 | Benzylamine coupling reaction performance of functional 3D COFs.**
**a** COF-catalyzed photooxidative coupling of benzylamines and their corresponding conversion rates. Reaction conditions: photocatalyst (8 mg), benzylamine (0.1 mmol), LED lamp (420–1100 nm, 15 W), CH$_3$CN (2 mL), O$_2$ (1 atm), 11 h. PXRD patterns (**b**) and FT-IR spectra (**c**) of COF-300-D-F before and after photocatalytic benzylamine coupling. **d** Five catalytic cycles of benzylamine coupling with COF-300-D-F as the photocatalyst and benzylamine as the substrate. **e** In situ DRIFTS experiment of COF-300-D-F collected at time-dependent photoirradiation under an O$_2$ environment. **f** The quenching experiments of COF-300-D-F for photocatalytic benzylamine oxidative coupling reactions. **g** Proposed coupling mechanism of benzylamine. Experiments (**a**–**f**) were repeated at least three times independently with similar results. a.u. indicates the arbitrary units in (**b**, **c**, **e**). Source data are provided as a Source data file.

## Structural characterization

Powder X-ray diffraction (PXRD) patterns were collected on a multi-purpose X-ray diffractometer (DX-2700BH, Haoyuan Instrument, China) using Cu Kα radiation (40 kV). Small-angle X-ray scattering (SAXS) measurements were performed on a Rigaku NanoPix instrument equipped with a rotating Cu anode and a confocal MaxFlux mirror (focal length 4000 mm), operating at 45 kV and 66 mA. Data were acquired in transmission mode using a 2D detector; ethanol-impregnated samples were mounted between two Kapton films (thickness: 1.5 mm). Elemental analysis (EA) was conducted on an Elementar Vario EL Cube analyser. Fourier-transform infrared (FT-IR) spectra were recorded on a Nicolet iS50 spectrometer (Thermo Fisher Scientific, USA) equipped with a liquid-nitrogen-cooled MCT detector. Solid-state $^{13}C$ cross-polarization magic angle spinning (CP/MAS) NMR spectra were acquired on a Bruker Avance III HD 400 MHz spectrometer using a 3.2 mm double-resonance MAS probe. Scanning electron microscopy (SEM) images were obtained with a Hitachi Regulus8220 field-emission microscope (Japan). Energy-dispersive spectroscopy (EDS) elemental mapping was performed on an FEI Titan Themis 60–300 operated at 200 kV. Electrochemical measurements, including photocurrent responses and Mott–Schottky plots, were performed using a Gamry Reference 600 workstation. Thermogravimetric analysis (TGA) was carried out on a Netzsch TG209F3 system; activated samples were placed in $Al_2O_3$ pans and heated from 35 to 800 °C at 10 °C min$^{-1}$ under $N_2$ flow (20 mL min$^{-1}$). Nitrogen adsorption–desorption isotherms were measured at 77 K using a Quantachrome Autosorb iQ$_2$ volumetric instrument. Surface areas were calculated by the Brunauer–Emmett–Teller (BET) method, and pore volumes were determined at $P/P_0 = 0.95$. UV-Vis diffuse reflectance spectra (200–800 nm) were recorded on a Shimadzu UV-3600i Plus spectrophotometer. Photoluminescence (PL) spectra were obtained with an Edinburgh FLS1000 fluorescence spectrometer. Electron paramagnetic resonance (EPR) spectra were recorded on a Bruker EPR EMX Plus spectrometer operating at 9.8 GHz (microwave power: 1 mW). Laser scanning confocal microscopy (LSCM) images were acquired on a Leica Stellaris 5 microscope. Photocatalytic benzylamine oxidative coupling was monitored by gas chromatography–mass spectrometry (GC-MS, Clarus 690, PerkinElmer) and gas chromatography (GC-2010 Pro, Shimadzu).

## Synthesis of COF-300

COF-300 was synthesized using a one-step solvothermal method[66,67]. A Pyrex tube measuring 10 × 8 mm (o.d × i.d) was charged with terephthalaldehyde (BDA, 13.4 mg, 0.1 mmol), tetrakis(4-aminophenyl) methane (TAM, 19.0 mg, 0.05 mmol), 1 mL of dioxane, and 0.2 mL of 6 M aqueous acetic acid. The solution was well mixed after 5 minutes ultrasnication and then degassed through three consecutive freeze–pump–thaw cycles and flame-sealed under vacuum. Upon sealing, the tube length was reduced to 18–20 cm. The reaction was heated at 120 °C for 72 h, yielding a yellow solid at the bottom of the tube. This solid was isolated by filtration, washed with dioxane and tetrahydrofuran, and the resulting powder was extracted with tetrahydrofuran for 24 h using a Soxhlet extractor. Finally, the solid was dried by supercritical $CO_2$ three times to obtain a yellow powder. Yield: 26 mg, 79.9%. The synthesis diagram is shown in Supplementary Fig. 2.

## Synthesis of COF-300-D

COF-300-D (D = 5%, 10%, 15%, 20%) was synthesized by a one-step solvothermal method and modified on the basis of COF-300. A Pyrex tube measuring 10 × 8 mm (o.d × i.d) was charged with BDA (13.4 mg, 0.1 mmol), TAM, and N,N-bis(4-aminophenyl)benzene-1,4-diamine (TAPA), 1 mL of dioxane, and 0.2 mL of 6 M aqueous acetic acid. The amounts of TAM and TAPA used for the synthesis of COF-300-D are shown in Supplementary Table 3. The subsequent operations are identical to those of the COF-300. The elemental analysis is shown in

Supplementary Table 4. The synthesis diagram is shown in Supplementary Fig. 3.

## Synthesis of COF-300-D-R

COF-300-D-R (D = 15%; R=F, Tz, OMe, OH) was synthesized by a one-step solvothermal method and modified based on COF-300-D[41]. A Pyrex tube measuring 10 × 8 mm (o.d × i.d) was charged with BDA, TAM (16.8 mg), TAPA (2.3 mg), and a series of different side chains of BDA (2,5-difluoroterephthalaldehyde, BDA-F; 2,1,3-benzothiadiazole-4,7-dicarboxaldehyde, BDA-Tz; 2,5-dimethoxy-1,4-benzenedicarboxaldehyde, BDA-OMe; and 2, 5- dihydroxy-1, 4- benzenedicarboxaldehyde, BDA-OH), 0.3 mL of N, N-dimethylacetamide (DMA), 0.7 mL of 1,2-dichlorobenzene (DCB) and 0.2 mL of 3 M aqueous acetic acid. The amount of different connector qualities used for the synthesis of COF-300-D-R is shown in Supplementary Table 5. The subsequent operations are the same as those of COF-300. The synthesis diagram is shown in Supplementary Fig. 24.

## Synthesis of COF-320

COF-320 was synthesized via a one-step solvothermal method[26]. In a typical procedure, a Pyrex tube (10 × 8 mm, o.d. × i.d.) was charged with TAM (20 mg, 0.05 mmol), 4,4′-biphenyldialdehyde (BPDA, 21 mg, 0.1 mmol), 1,4-dioxane (0.25 mL), pyridine (0.15 mL), aniline (0.06 mL), and 0.2 mL of 6 M aqueous acetic acid. The mixture was ultrasonicated for 5 min to ensure homogeneity. Subsequently, the reaction mixture was degassed through three freeze–pump–thaw cycles and flame-sealed under vacuum. After sealing, the tube was shortened to a length of 18–20 cm. The reaction was heated at 85 °C for 72 h, resulting in the formation of a yellow solid at the bottom of the tube. The product was collected by filtration, washed with dioxane and tetrahydrofuran, and then subjected to Soxhlet extraction with tetrahydrofuran for 24 h. Finally, the solid was dried using supercritical $CO_2$ three times to yield a yellow powder. A schematic representation of the synthesis is provided in Supplementary Fig. 28.

## Synthesis of COF-320-D-15%

COF-320-D-15% was synthesized by a one-step solvothermal method and modified on the basis of COF-320. A Pyrex tube measuring 10 × 8 mm (o.d. × i.d.) was charged with TAM (16.4 mg, 0.043 mmol), TAPA (2.3 mg, 0.008 mmol), 4,4′-biphenyldialdehyde (BPDA) (21 mg, 0.1 mmol), 1,4-dioxane (0.25 mL), and pyridine (0.15 mL), aniline (0.06 mL), and 0.2 mL of 6 M aqueous acetic acid water. The subsequent operations are the same as those of COF-320. The synthesis diagram is shown in Supplementary Fig. 29.

## Synthesis of COF-320-D-F

COF-320-D-F was synthesized by a one-step solvothermal method and modified on the basis of COF-320-D-15%. A Pyrex tube measuring 10 × 8 mm (o.d. × i.d.) was charged with TAM (16.4 mg, 0.043 mmol), TAPA (2.3 mg, 0.008 mmol), 4,4′-biphenyldialdehyde (BPDA) (10.5 mg, 0.05 mmol), [1,1′-biphenyl]-4,4′-dicarboxaldehyde, 2,2′,3,3′,5,5′,6,6′-octafluoro- (BPDA-F) (17.7 mg, 0.05 mmol), 1,4-dioxane (0.25 mL), and pyridine (0.15 mL), aniline (0.06 mL) and 0.2 mL of 6 M aqueous acetic acid water. The subsequent operations are the same as those of COF-320-D-15%. The synthesis diagram is shown in Supplementary Fig. 30.

## Laser scanning confocal microscopy (LSCM)

To investigate the spatial distribution of TAPA linkers within COF-300-D crystals, LSCM was employed. Single-crystal samples approximately 6 µm in size were prepared following a reported procedure[24]. All measurements were performed on a Leica Stellaris 5 microscope. A 405 nm laser was used as the excitation source to induce fluorescence from the TAPA linkers within the COF crystals. Imaging was conducted in a direction perpendicular to the glass slide, with optical sectioning across 70 planes at progressively increasing depths. Each section

exhibited red fluorescence, indicating a homogeneous distribution of TAPA linkers throughout the entire COF-300-D crystal. Importantly, identical instrument settings were maintained during the imaging of all samples to ensure consistency across measurements.

## Liquid nuclear magnetic resonance spectroscopy

Liquid $^1H$ and $^{13}C$ NMR spectra of the COF powder were recorded after the digestion procedure (Bruker Avance III HD 400 MHz). Approximately 3–4 mg of fully activated COF powder was dissolved in a mixture of 50 μL DCl (38 wt% in $D_2O$) and 500 μL DMSO-$d_6$, followed by sonication for 30 min until a clear solution formed. The homogeneous mixture solution was used directly for $^1H$ and $^{13}C$ NMR measurements.

The actual molar fraction of TAPA ($f_{TAPA}$) in the COF-300-D series was determined via $^1H$ NMR analysis of the digested samples (Supplementary Figs. 16-19). The well-resolved signal from a designated proton of the BDA linker ($H_1$, $\delta = 10.17$ ppm) was used as an internal standard. This is justified because the molar ratio of (TAM + TAPA) to BDA is fixed at 1:2 in the COF structure, ensuring a constant reference point.

The calculation was based on the integrated area of the overlapping signals $I(H_{3,4,6})$, which includes H-3 and H-4 of TAM and H-6 of TAPA ($\delta$ 7.37–7.44 ppm). The mole fraction $f_{TAPA}$ was calculated using the following equation derived from proton count conservation:

$$f_{TAPA} = \frac{\left[ k \times \frac{I(H_{3,4,6})}{I(H_1)} \right] - n}{m - n} \times 100\% \tag{1}$$

Where $k$ is the number of protons that contribute to $I(H_1)$ from the BDA unit corresponding to one TAM unit in COF-300, and here is a constant 4. $I(H_{3,4,6})$ is the integrated area of the overlapping peaks (H-3, H-4 of TAM and H-6 of TAPA), $I(H_1)$ is the integrated area of the peak (H-1), $n$ is the number of protons from one TAM unit that contribute to $I(H_{3,4,6})$, and here is a constant 16. $m$ is the number of protons from one TAPA unit that contribute to $I(H_{3,4,6})$; here, it is a constant of 6.

The equation is derived as follows: The ratio $I(H_{3,4,6})/I(H_1)$ is proportional to the number of protons in the overlap region per BDA unit. Therefore:

$$k \times \frac{I(H_{3,4,6})}{I(H_1)} = n(1 - f_{TAPA}) + m \times f_{TAPA} \tag{2}$$

## $H_2O_2$ detection

The amount of $H_2O_2$ was determined using the iodometry method[68]. In a typical procedure, 0.5 mL of potassium iodide (KI) solution (0.4 mol $L^{-1}$) and 0.5 mL of potassium hydrogen phthalate ($C_8H_5KO_4$) solution (0.1 mol $L^{-1}$) were added to 0.1 mL of diluted solution and allowed to react for 30 min. $H_2O_2$ was reacted with $I^-$ under acidic conditions to form $I_3^-$ ($H_2O_2 + 3I^- + 2H^+ \rightarrow I_3^- + 2H_2O$), which exhibits strong absorption at approximately 350 nm as measured by UV–vis spectroscopy. The total amount of $H_2O_2$ generated during the reaction was quantified based on this method.

## $H_2O_2$ photosynthesis

Experiments on $H_2O_2$ photosynthesis were conducted following a procedure adapted from the literature[69]. Briefly, 1 mg of photocatalyst was added to a mixture of deionized water (18 mL) and benzyl alcohol (2 mL) in a 100 mL quartz beaker (maximum diameter, $\varphi$ 40 mm; open to air). The mixture was ultrasonicated for 3 min to disperse the catalyst, then stirred in the dark for 10 min to establish adsorption–desorption equilibrium under ambient conditions. Irradiation was performed using a 300 W Xe lamp (CEL-HXF300-(T3), Beijing China Education AU-Light Technology Co., Ltd.) equipped with a cutoff filter ($\lambda > 420$ nm), with a light power density of $500 \pm 10$ mW $cm^{-2}$ as measured by an automatic optical power meter (CEL-NP2000-2(10)A, Beijing China Education AU-Light Technology Co., Ltd.). The reaction temperature was maintained

at $25 \pm 1$ °C under continuous magnetic stirring. At designated time intervals, aliquots were collected and filtered through a 0.22 μm filter to separate the solid. The $H_2O_2$ concentration in the filtrate was subsequently determined.

To assess the influence of Ar on photocatalytic $H_2O_2$ production, Ar gas was continuously bubbled into the reaction solution in the dark for 30 min prior to irradiation. Photocatalysis was then carried out under a continuous Ar flow throughout the reaction. To probe the mechanism of $H_2O_2$ generation, a series of scavenger experiments was performed by separately adding benzoquinone (BQ, 10 mM), tert-butyl alcohol (TBA, 10 mM), and $AgNO_3$ (10 mM) to the reaction system, which serve as scavengers for superoxide radicals ($\cdot O_2^-$), hydroxyl radicals ($\cdot OH$), and electrons ($e^-$), respectively. Long-term stability tests were conducted under optimized experimental conditions– including enhanced sealing and thermal management–to minimize solvent evaporation.

## Measurement of apparent quantum yield (AQY)

The AQY was measured following a previously reported procedure[70]. Irradiation of the photocatalytic experiment was provided by a 300 W Xe lamp equipped with bandpass filters at 400, 420, 450, 475, and 520 nm. 20 mg COF powder was dispersed in the mixed solution of 18 mL deionized water and 2 mL benzyl alcohol, followed by ultrasonic treatment to fully disperse it. The AQY was then calculated using the equation:

$$AQY = \frac{2 \times H_2O_2 \ formed \ (mol)}{the \ number \ of \ incident \ photons \ (mol)} \times 100\% \tag{3}$$

$$AQY = \frac{\left( M_{H_2O_2} \times N_A \times h \times c \right) \times 2}{S \times P \times T \times \lambda} \times 100\% \tag{4}$$

Where $M$ is the yield of $H_2O_2$ (mol), $N_A$ is Avogadro's constant ($6.022 \times 10^{23}$ $mol^{-1}$), $h$ is the Planck constant ($6.626 \times 10^{-34}$ J·s), $c$ is the speed of light ($3 \times 10^8$ $ms^{-1}$), $S$ is the irradiation area ($cm^2$), $P$ is the intensity of irradiation light (W $cm^{-2}$), $T$ is the photoreaction time (s), $\lambda$ is the wavelength of the monochromatic light (m).

## Photoelectrochemical measurements

Photoelectrochemical measurements were performed following a procedure adapted from the literature[69]. Mott–Schottky plots and photocurrent responses were recorded using an electrochemical workstation (Gamry Reference 600, USA). A 300 W Xe lamp ($\lambda > 420$ nm; CEL-HXF300-T3, Beijing Zhongjiao Jinyuan Technology Co., Ltd.) served as the light source, and an aqueous 0.5 M $Na_2SO_4$ solution was used as the supporting electrolyte for all photocurrent measurements. Working electrodes were prepared as follows: 5 mg of photocatalyst was dispersed in a mixture of 450 μL of ethanol and 50 μL of Nafion by sonication for 20 min. Subsequently, 50 μL of the resulting slurry was evenly drop-cast onto an indium tin oxide (ITO)-coated glass substrate ($1 \times 1$ $cm^2$), corresponding to a catalyst loading of 0.5 mg $cm^{-2}$, and dried at room temperature.

All electrochemical measurements were conducted in a standard three-electrode configuration, with a platinum wire as the counter electrode and an Ag/AgCl electrode as the reference. Prior to each measurement, the Ag/AgCl reference electrode was calibrated against a platinum wire in a hydrogen-saturated electrolyte. Potentials for the oxygen reduction reaction (ORR) are reported relative to the reversible hydrogen electrode (RHE). Fresh 0.5 M $Na_2SO_4$ electrolyte (pH = $6.8 \pm 0.2$) was used for all experiments. The solution exhibited a resistance of $14.9 \pm 0.5$ $\Omega$, and all measurements were corrected for the iR drop in real time by means of automatic compensation.

Mott–Schottky measurements were carried out over a potential range of −1 to 1 V at frequencies of 1000, 1500, and 2000 Hz.

Photocurrent responses were recorded under intermittent illumination from a 300 W Xe lamp. Transient photocurrent profiles were collected over five on–off cycles, each lasting 40 s, at an initial voltage of 0 V (sampling interval: 0.1 s; sensitivity: $1 \times 10^{-5}$ AV$^{-1}$).

### In-situ diffuse reflectance infrared Fourier transform spectroscopy (DRIFTS)

DRIFTS was conducted following established procedures[59,64]. Spectra were collected using a Nicolet iS-50 Fourier-transform infrared spectrometer coupled with a Harrick diffuse reflectance accessory. Powder samples were loaded into an environmental cell (Harrick Scientific Products, Inc.), which is optimized for diffuse reflectance measurements on highly scattering materials. The cell was sealed with two ZnSe windows to enable gas-phase control during analysis.

### Electron paramagnetic resonance (EPR) measurements

Spin-trapping EPR measurements were performed using a Bruker EMXplus-6/1 spectrometer operating at X-band (9.4 GHz). 5,5-Dimethyl-1-pyrroline N-oxide (DMPO) was employed as the spin-trapping agent to detect superoxide radicals ($\cdot O_2^-$) or hydroxyl radicals ($\cdot OH$). Specifically, 2 mg of the COF powder was dispersed in 500 μL of either water or a methanol/water mixed solution (9/1 v/v) containing DMPO (0.1 mmol) inside a Pyrex tube. The tube was subsequently sealed with a rubber septum. A 300 W Xe lamp equipped with a 420 nm cut-off filter was used for irradiation.

### Photocatalytic oxidative coupling of benzylamines

Photocatalytic experiments were conducted as follows: 8 mg of various COF catalysts were added to a quartz reactor containing a mixture of 2 mL acetonitrile and 0.1 mmol benzylamine[71]. After continuously bubbling oxygen through the solution for at least 1 min, the reactor was sealed and equipped with an $O_2$ balloon. Throughout the reaction, the temperature was maintained at 298 K using a circulating water-cooling system, and the reactor was irradiated under a 15 W LED lamp (Shanghai Shanshi Technology Co., Ltd., Shanghai, China). After the reaction, a certain amount of solution was collected and filtered through a 0.22 μm syringe filter to remove catalyst particles, and then analyzed by gas chromatography.

### Theoretical calculations

The density functional theory (DFT) calculations were performed using the Vienna ab initio simulation package (VASP 5.4.4)[72,73]. The interaction between core and valence electrons was described using the spin-polarized projector augmented wave (PAW) method[74], while the electron exchange-correlation effects were treated within the generalized gradient approximation (GGA) using the Perdew-Burke-Ernzerhof (PBE) functional[75]. The electronic wave functions were expanded in a plane-wave basis set with an energy cutoff of 450 eV. Structural optimization was performed with convergence thresholds of $10^{-5}$ eV for the electronic self-consistent loop and 0.02 eV/Å for the residual forces. A $1 \times 1 \times 1$ K-point mesh was used for the electronic structure calculations. In addition, the Van der Waals interactions were considered using the DFT-D3 correction scheme[76]. Atomic charge distribution was evaluated based on the Bader atom-in-molecule (AIM) approach[77]. Part of the electronic structure analysis was conducted with the assistance of the VASPKIT post-processing package[78].

To quantify the interaction strength between the adsorbate and the substrate, the binding energy was defined as:

$$E_b = E_{total} - E_{mol} - E_{sub} \qquad (5)$$

where $E_{total}$, $E_{sub}$, and $E_{mol}$ represent the total energy of the adsorption system, the pristine substrate, and the isolated molecule in the gas phase, respectively.

## Data availability

All data are available in the main text or the Supplementary Information. Source data are provided with this paper.

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

## Acknowledgements

This work was financially sponsored by the National Key Research and Development Program of China (grant no. 2024YFE0201200 [C.C.]), the National Natural Science Foundation of China (grant nos. 52525311 [C.C.], 52373148 [C.C.], 82302224 [X.H.X.], 52273269 [S.L.], 82561160101 [C.C.]), the Sichuan Science and Technology Program (No. 2024YFHZ0270 [X.H.X.]), Sichuan University Interdisciplinary Innovation Fund [X.H.X.], and the Support from State Key Laboratory of Synergistic Chem-Bio Synthesis (sklscbs202576 [C.C.]), and the State Key Laboratory of Advanced Polymer Materials (Grant No. sklpme2024-2-14 [X.H.X.]). Dr. Mi Zhou thanks the support from the Opening Project of the Sichuan Provincial Engineering Research Center of Functional Development and Application of High Performance Special Textile Materials (Chengdu Textile College, Project Number: 2024FDAST-C08). We would like to thank Dr. Peng Wu & Dr. Yanying Wang of the Analytical & Testing Center, Sichuan University, for their assistance on steady/transient fluorescence. We also thank the scientists from Wuhan University for their assistance with SAXS measurements.

## Author contributions

X.H.X., T.T.D., C.C., and S.L. proposed the idea and designed the experiments. T.T.D., J.N.Y., M.C.G., M.Z., M.X., and W.C.X. performed the experiments, characterization, and results analysis. T.T.D. and L.C. assisted with the figure production and experiment design. T.T.D., C.C., and L.C. designed and conducted the theoretical calculation. T.T.D., X.H.X., S.L., and C.C. wrote and edited the manuscript. X.H.X., X.C.R., S.L., and C.C. supervised the whole project. All authors discussed the results and commented on the manuscript.

## Competing interests

The authors declare no competing interests.
