## [Transparent Peer Review file · Nature Communications]

Defective Three-dimensional Covalent Organic Frameworks for Enhanced Hydrogen Peroxide Photosynthesis and Organic Transformation

Corresponding Author: Professor Chong Cheng

Version 0:

Reviewer comments:

Reviewer #1

(Remarks to the Author)

The authors successfully presented a knot-connectivity reduction strategy for introducing planar light-harvesting units into 3D COFs by partially replacing tetrahedral knots with trigonal planar ligands, while preserving the underlying network. Systematically replacing tetrahedral knots with trigonal planar ligands and modifying linear linkers with electron-withdrawing/donating groups enabled simultaneous enhancement of light absorption and precise electronic tuning of 3D donor-acceptor structures. The optimized 3D COFs with planar ligands, induced defects, and fluorine functional groups (COF-300-D-F) appeared to possess high photocatalytic abilities for H₂O₂ synthesis and organic transformation, which was particularly impressive. However, the following issues must be addressed before this manuscript can be accepted.

1. The photocatalytic H₂O₂ production rates of the COF-300-D series increased first and then decreased with the increase of TAPA in the range of 0%~20%, among which COF-300-D-15% showed the highest rate of 1.74 mmol g⁻¹h⁻¹ (Fig. 4a and Supplementary Fig. 64). Why is the amount of H₂O₂ produced not increasing monotonically?
2. The authors demonstrated that COF-300-D-F had a long-term photocatalytic stability (as shown in Fig. 4g) and the final concentration of H₂O₂ reached 0.95 wt%. To properly demonstrate the high stability of COF-300-D-F, characterization of its chemical structure and crystal structure after long-term use is essential. Furthermore, it is also important to consider the maximum hydrogen peroxide concentration at which this COF can maintain structural stability.
3. Regarding Fig. 4g, why does the amount of H₂O₂ produced per 24 hours increase towards the end of the experiment?
4. In the photocatalytic H₂O₂ production, the authors used benzyl alcohol as a hole scavenger, which diminished the novelty of this work. First, the authors should clearly state that the impressive H₂O₂ production rates described in the abstract, introduction, and conclusion are achieved using a hole scavenger. Second, benzyl alcohol and its oxidized products may be oxidized by hydrogen peroxide and reactive oxygen species, which would undermine the practicality of this work. This could be clarified, for example, by analyzing the solution after long-term measurements (Fig. 4g). Finally, is it possible to produce H₂O₂ without these sacrificial reagents, and if not, how can this be achieved?

Minor issues

5. Is 20% the upper limit for TAPA that can be incorporated into COF-300-D?
6. Details of Pawley refinements should be described in the Supplementary Methods.
7. The space between 0 and 5 in Supplementary Fig. 40 measurement wavelength ($\lambda = 1.54056 \text{ \AA}$) is unnecessary.
8. The author should add the redox potential for the oxidation of benzyl alcohol to benzyl aldehyde to Fig. 3b.
9. Fig. 3d discusses the luminescence intensity of each compound, but is there any information on the photoluminescence quantum yield?

Reviewer #2

(Remarks to the Author)

The manuscript describes a strategy to enhance the photocatalytic activity of 3D covalent organic frameworks (COFs) by partially replacing tetrahedral nodes in COF-300 with planar tri-connected ligands (TAPA) and replacing linear 2-c linkers with electron-withdrawing or -donating groups. This approach tailors light absorption, charge separation, and overall

catalytic efficiency. The authors demonstrate very high rates of H₂O₂ production, excellent stability over extended operation, and good performance in benzylamine coupling reactions.

The work is well presented and the experiments are solid. The catalytic results are particularly impressive, showing clear advantages over comparable systems in the literature. The characterization is thorough and supports the conclusions drawn. I believe the manuscript is suitable for Nature Communications, although some points could be clarified to improve the overall message.

My main comment concerns how the strategy is framed. The term knot-connectivity reduction strategy gives the impression of precise control over node connectivity. However, what the study really shows is that using mixed ligands with varying connectivity introduces defects in a controlled way up to a point, but with increasing disorder as the proportion of tri-connected ligands rises (PXRD and SAXS). The authors could describe this in a more balanced way, highlighting both the benefits and the structural limits of the approach:

1. I suggest the authors reconsider the way they describe their strategy. Rather than "knot-connectivity reduction," it might be clearer to present it as a mixed-ligand or defect engineering approach that allows introduction of planar units and tuning of electronic properties, while acknowledging the gradual loss of order at higher TAPA content. Similar strategies are followed in metal-organic frameworks and are often referred to as ligand-truncation.

2. It would strengthen the paper to include a short discussion on the limits of this approach. For example, what is the maximum amount of low-connectivity ligand the structure can tolerate before crystallinity is lost or the dia net is not assembled? This would help readers better understand how the method could be applied to other COF systems.

I recommend minor revision. The study is of high quality, and the suggested clarifications would help sharpen the main message. The manuscript is generally clear and well written, though a few sections could benefit from slightly more concise language.

Reviewer #3

(Remarks to the Author)

In this study, the authors introduce a knot-connectivity reduction strategy to construct 3D COFs for hydrogen peroxide photocatalysis and photocatalytic benzylamine coupling reactions. I highly support its publication in Nature Communications after sufficient revisions with the following considerations:

1. The authors did not determine the actual structure of the series of 3D COFs, especially without considering the impact of the node replacement strategy on the interpenetration structure of dia topology network. Therefore, with the increase of defect concentration in COFs, does the full width at half maximum (FWHM) of diffraction peaks in PXRD spectra show significant changes? Is the distribution of these defects uniform? What about the pore size distribution in the COF-300-D series, and is it allowed for benzylamine to enter the pores and contact with catalytic sites?

2. The authors should consider the possibility of BDA monomers being connected to TAPA monomers at both ends in the COF-300-D series, rather than the idealized structure of one end TAPA and the other end TAM, especially when the concentration of TAPA increases.

3. The ¹H and ¹³C NMR did not show significant differences in the Supplementary Figures. How to calculate Supplementary Fig. 35 based on ¹H NMR? More details need to be provided.

4. Is this still a good strategy for the quantitative ratio of COF-320 series? And how about the photocatalytic performance of COF-320 series in producing hydrogen peroxide?

5. Why is the fluorescence emission peak of COF-300 in the Fig. 3d around 520 nm, while in Supplementary Fig. 60 the fluorescence emission peak of the three-dimensional fluorescence spectrum is around 600 nm? Does the lack of normalization of fluorescence intensity in Fig. 3d indicate a difference in the relative fluorescence intensity of these COFs? What are the excitation and emission wavelengths corresponding to the fluorescence lifetime in Fig. 3e?

6. Is there a problem of phase separation or purity at higher doping ratios (over 20%)? Why does the catalytic efficiency decrease when the defect concentration is higher (20%) in Fig. 4a? How does defect concentration affect the mass transfer rate in photocatalytic reactions?

7. What is the photocatalytic efficiency of COF-300-D series for hydrogen peroxide production in pure water and different hole scavengers?

8. Theoretical calculations should use periodic 3D COFs structures rather than local molecular segments, because it is necessary to take into account the stacking effect and charge transfer between structural units caused by the interweaving of topological structures.

Version 1:

Reviewer comments:

Reviewer #1

(Remarks to the Author)

Thank you for responding to my comments. The authors' responses are partially incomplete; therefore, I could reconsider publishing this manuscript if they fully address the comments below.

About Question (1),

The authors mentioned that the non-monotonic trend in H₂O₂ production is related to the competition between two opposing effects (the introduction of beneficial light-harvesting units and the gradual disruption of the long-range order and crystallinity of the COF framework). Are there any results or previous studies that support this phenomenon?

About Question (2),

The authors should add an explanation of structural stability after 120 hours in the revised manuscript, in the same manner as they did after 96 hours.

About Question (3),

Although the authors carefully improved their experimental method, Figure 4g still shows an increase in H₂O₂ production per unit time after 24 hours; therefore, the authors should explain this phenomenon and its underlying mechanism.

About Question (4),

Thank you for agreeing that the use of sacrificial agents diminishes the practicality of this research.

About Question (6),

Although the specific parameters of Pawley refinements were added to the supplementary information, the standard deviation of the lattice constants obtained by Pawley refinements should also be included.

Reviewer #2

(Remarks to the Author)

The authors have provided a thorough and convincing response to all my comments. I am satisfied with the changes and consider the paper suitable for publication in Nature Communications in its current form.

Reviewer #3

(Remarks to the Author)

The authors have done a great job in addressing my previous concerns. Thus, I highly recommend it for publication in Nature Communications as it is.

Version 2:

Reviewer comments:

Reviewer #1

(Remarks to the Author)

Thank you for responding to my comments. Your answer pointed out the use of sacrificial reagents and the structural collapse after 120 hours of operation (even at low H₂O₂ concentrations), and I believe these findings indicate the fatal limitations of this material as a catalyst for hydrogen peroxide production. Given that, as you mentioned, there are many similar studies on catalysts for hydrogen peroxide production, I recommend publishing your manuscript in a more specialized journal, such as Communications Materials.

Point-by-point response to the detailed comments by reviewers of “*Molecular Engineering of Defective 3D Covalent Organic Frameworks for Enhanced Photocatalytic H₂O₂ Synthesis and Organic Transformation*” with manuscript ID: NCOMMS-25-38777A.

REVIEWER COMMENTS

Reviewer #1 (Remarks to the Author):

“The authors successfully presented a knot-connectivity reduction strategy for introducing planar light-harvesting units into 3D COFs by partially replacing tetrahedral knots with trigonal planar ligands, while preserving the underlying network. Systematically replacing tetrahedral knots with trigonal planar ligands and modifying linear linkers with electron-withdrawing/donating groups enabled simultaneous enhancement of light absorption and precise electronic tuning of 3D donor-acceptor structures. The optimized 3D COFs with planar ligands, induced defects, and fluorine functional groups (COF-300-D-F) appeared to possess high photocatalytic abilities for H₂O₂ synthesis and organic transformation, which was particularly impressive. However, the following issues must be addressed before this manuscript can be accepted.”

Response to the general comment:

We sincerely appreciate your recognition of our investigations on the knot-connectivity reduction strategy. Based on your comments and the suggestions from other reviewers, we have conducted more systematic experiments and refined the content throughout the manuscript. All necessary data have been added to support our claims, and we have thoroughly addressed all questions and concerns in the revised manuscript and supplementary information. Therefore, we believe that the quality of this paper has been significantly enhanced. We hope you will agree with this assessment, and we thank you once again for your considerable efforts.

(1) *The photocatalytic H₂O₂ production rates of the COF-300-D series increased first and then decreased with the increase of TAPA in the range of 0%~20%, among which COF-300-D-15% showed*

the highest rate of 1.74 mmol g⁻¹h⁻¹ (Fig. 4a and Supplementary Fig. 64). Why is the amount of H₂O₂ produced not increasing monotonically?

Response to comment:

Thank you for your attentive review and valuable feedback. The non-monotonic trend in H₂O₂ production through photocatalysis is indeed an interesting result of our research. We are very pleased to have the opportunity to elaborate on the underlying mechanism in detail. We believe that as the amount of TAPA increases, this phenomenon is caused by the competition between two opposing effects: the introduction of beneficial light-harvesting units and the gradual disruption of the long-range order and crystallinity of the COF framework. The optimal performance at 15% TAPA represents the balance between these two competing factors. At this concentration, the positive effects of enhanced charge separation and the introduction of new active sites maximize the photocatalytic output, while the negative impact of structural disorder is still minimized. Further increasing the TAPA content would disrupt this balance, as the adverse effects of crystallinity loss and framework disorder outweigh the benefits of introducing active sites, resulting in the observed decrease in activity. This structure-property relationship underscores the importance of precise component control in multivariate COFs for enhancing photocatalytic performance. Thank you again for your insightful and valuable suggestions.

(2) The authors demonstrated that COF-300-D-F had a long-term photocatalytic stability (as shown in Fig. 4g) and the final concentration of H₂O₂ reached 0.95 wt%. To properly demonstrate the high stability of COF-300-D-F, characterization of its chemical structure and crystal structure after long-term use is essential. Furthermore, it is also important to consider the maximum hydrogen peroxide concentration at which this COF can maintain structural stability.

Response to comment:

Thanks for your important and helpful comments on improving the quality of our manuscript. We acknowledge your suggestion to characterize the chemical and crystal structures of COF-300-D-F after long-term use to verify its high stability. We also tested the maximum hydrogen peroxide concentration at which COF-300-D-F maintains its structural stability. The experimental results showed that sharp

diffraction peaks could be observed in the PXRD pattern after a 96-hour photocatalytic reaction, indicating that the crystallinity and structural integrity were still maintained. However, the crystalline structure collapsed after 120 hours, as evidenced by its amorphous diffraction pattern. However, obvious signals of the imine linkages ($C=N$, 1624 cm^{-1}) and unbroken spindinle particles could be observed from the IR spectra and SEM images of COF-300-D-F after a 120 h reaction and even up to 144 h, which suggested that the long-range ordered structure partially decomposed into a short-range cross-linking structure of the building units. Moreover, after a 96-hour photocatalysis reaction of COF-300-D-F, a 0.10% concentration of hydrogen peroxide solution was collected. The details can be found in the revised manuscript and the revised supplementary information, as follows:

Page 14 in the revised manuscript: “It is worth noting that this unique synthesis strategy provides COF-300-D-F with remarkable long-term photocatalytic structural stability and performance stability, which can run continuously for more than 96 hours without obvious decline. The final concentration of H_2O_2 reaches 0.10 wt%. It can be used for daily applications, such as oral cleaning directly (Fig. 4g and Supplementary Fig. 79). The characterization of PXRD, FTIR, and SEM after 96 hours of photocatalytic operation showed that the structural integrity and morphological stability of COF-300-D-F were maintained after long-term photocatalytic production of H_2O_2 . No obvious degradation was observed compared with that before photocatalytic reaction (Supplementary Fig. 80).”

Fig. 4 H_2O_2 evolution performance of functional 3D COFs. **g**, Continuous and stable production of H_2O_2 over 96 hours of COF-300-D-F.

Supplementary Fig. 79 Schematic diagram of continuous and stable production of H₂O₂ over 96 hours and directly obtaining H₂O₂ solution.

Supplementary Fig. 80 The long-term photocatalytic reaction of COF-300-D-F to produce H₂O₂ at different reaction times was characterized by PXRD (a), FTIR (b), and SEM (c).

(3) *Regarding Fig. 4g, why does the amount of H₂O₂ produced per 24 hours increase towards the end of the experiment?*

Response to comment:

We are deeply grateful for your astute observation regarding the anomalous increase in the H₂O₂ production rate during the long-term test in our original manuscript. Your comment prompted a thorough re-evaluation of our experimental approach, leading to significant methodological improvements and a more accurate representation of the catalyst's performance. Upon meticulous investigation, we identified that subtle but cumulative solvent evaporation over the extreme duration of the experiment (200 hours), even from a nominally closed system, was the primary factor leading to the observed artifact. The gradual concentration of reactants and the catalyst artificially inflated the apparent reaction rate in the later stages. To unequivocally resolve this issue and acquire robust data, we have implemented a comprehensive set of optimized experimental controls for the long-term stability assessment: (1) Enhanced Sealing: The reactor assembly was rigorously sealed to minimize any potential vapor loss. (2) Optimized Thermal Management: The temperature of the cooling water was precisely lowered from 25°C to 20°C to better condense any volatilized solvent within the system. (3) Reduced Mechanical Agitation: The stirring speed was appropriately lowered to decrease the energy input and associated heating, further mitigating evaporation. (4) Minimized Sampling Impact: The number of sampling events was strictly minimized to maintain a constant system volume and composition. The results obtained under these stringently controlled conditions are now presented in the revised Fig. 4g. The data unambiguously demonstrate a stable H₂O₂ production rate over 96 hours, thereby confidently confirming the excellent operational stability of the COF-300-D-F catalyst, without interference from systemic artifacts.

We have updated the manuscript accordingly and sincerely apologize for the oversight in our initial experimental design. We believe that the rigorous process of addressing your concern has significantly strengthened the work. Your insightful feedback was invaluable in guiding these improvements, for which we are truly thankful. The details can be found in the revised manuscript and the revised supplementary information, as follows:

Fig. 4 H₂O₂ evolution performance of functional 3D COFs. **g**, Continuous and stable production of H₂O₂ over 96 hours of COF-300-D-F.

Page S72 in the revised supplementary information: “... The long-term stability test presented was performed with optimized experimental safeguards against solvent evaporation. These included enhanced sealing and thermal management.”

(4) In the photocatalytic H₂O₂ production, the authors used benzyl alcohol as a hole scavenger, which diminished the novelty of this work. First, the authors should clearly state that the impressive H₂O₂ production rates described in the abstract, introduction, and conclusion are achieved using a hole scavenger. Second, benzyl alcohol and its oxidized products may be oxidized by hydrogen peroxide and reactive oxygen species, which would undermine the practicality of this work. This could be clarified, for example, by analyzing the solution after long-term measurements (Fig. 4g). Finally, is it possible to produce H₂O₂ without these sacrificial reagents, and if not, how can this be achieved?

Response to comment:

We sincerely thank you for bringing this important point about the use of the hole scavenger to our attention. According to your suggestion, we revised the manuscript to enhance and clarify the objectives and results of our work.

As suggested, we have now explicitly stated in the abstract, introduction, and conclusion that the reported H₂O₂ production rates were measured in the presence of benzyl alcohol. The details can be found in the revised manuscript, as follows:

Page 2 in the revised manuscript: "... Experimental results and theoretical analysis revealed that the optimized 3D COFs with planar ligands induced defects and fluorine functional groups (COF-300-D-F) demonstrates unprecedented activity, achieving an H₂O₂ production rate of 19.09 mmol g⁻¹ h⁻¹ and apparent quantum yield (AQY) of 11.95% at 400 nm with benzyl alcohol (BnOH) as sacrificial agent, which represents the highest activities of 3D COF photocatalysts, even surpassing some inorganic catalytic materials."

Page 4 in the revised manuscript: "... In combination with partially replacing T_d knot (TAM) with TAPA and the use of BDA-F linker, the resulting COF-300-D-F demonstrates excellent performance, achieving an H₂O₂ production rate of 19.09 mmol g⁻¹ h⁻¹ at pH 3 with an AQY of 11.95% at 400 nm in the two-phase system of benzyl alcohol (BnOH) and deionized water."

Page 12 in the revised manuscript: "We initially selected benzyl alcohol (BnOH) as a hole sacrificial agent to evaluate the intrinsic activity of the COFs in the photocatalytic production of H₂O₂ under visible light ($\lambda > 420$ nm). The H₂O₂, as the O₂ reduction product, is dissolved in water, while the BnOH, as a hole scavenger, is oxidized into value-added benzaldehyde."

Page 20 in the revised manuscript: "The optimized COF-300-D-F exhibits exceptional performance for photocatalytic H₂O₂ production, achieving a rate of 19.09 mmol g⁻¹ h⁻¹ in a benzyl alcohol/water system. This process is paired with the valorization of the hole scavenger, transforming benzyl alcohol into value-added benzaldehyde, all while maintaining remarkable structural and operational stability for over 96 hours."

We acknowledge that the use of a sacrificial agent is a limitation for immediate practical application. However, we wish to emphasize that the primary goal of this work is to demonstrate a new concept by showing that our mixed-ligand strategy can successfully engineer the electronic structure of a 3D COF to enable selective photocatalytic H₂O₂ production. In this context, the use of a hole scavenger is a standard and necessary methodology for effectively isolating and evaluating the intrinsic potential of the material for the desired reduction reaction (H₂O₂ production) by rapidly consuming the photogenerated holes. This approach allows us to unequivocally attribute the performance differences to the structural and electronic modifications of the COFs, which is the core novelty of our work.

Additionally, we appreciate your insightful suggestions on analyzing the solution after the reaction. Therefore, we used ^1H NMR, ^{13}C NMR, and gas chromatography (GC) to analyze the solution in detail after a 96-hour-long-term photocatalytic reaction. The results confirmed that benzyl alcohol was oxidized to benzaldehyde. Crucially, and in contrast to what might be expected, we did not detect any benzoic acid. This new data has been added to the Supporting Information (Figure S81-S83). This finding indicates that under our reaction conditions, the further oxidation of benzaldehyde by H_2O_2 or reactive oxygen species is minimal. This reduces the complexity of subsequent separation to a certain extent and enhances the practicality of the system. The details can be found in the revised manuscript and the revised supplementary information, as follows:

Page 14 in the revised manuscript: “The solution extracted after the 96-hour reaction was further analyzed by gas chromatography (GC), ^1H NMR, and ^{13}C NMR (Supplementary Figs. 81-83). The results showed that BnOH was selectively oxidized to benzaldehyde, while benzoic acid was not detected.”

Supplementary Fig. 81 GC spectra of the reaction system solution of COF-300-D-F after 96 h of photocatalytic H_2O_2 production (benzyl alcohol as sacrificial agent) after extraction and the standard curves of benzyl alcohol, benzaldehyde, and benzoic acid. *, interior label.

Supplementary Fig. 82 ^1H NMR spectra in the $\text{DMSO-}d_6$ solvent of the reaction system solution of COF-300-D-F after 96 h of photocatalytic H_2O_2 production (benzyl alcohol as sacrificial agent) after extraction.

Supplementary Fig. 83 ^{13}C NMR spectra in the $\text{DMSO-}d_6$ solvent of the reaction system solution of COF-300-D-F after 96 h of photocatalytic H_2O_2 production (benzyl alcohol as sacrificial agent) after extraction.

We tested the photocatalytic yield of H₂O₂ for COF-300-D-F in pure water and different sacrificial agents, and added detailed discussion in the revised manuscript and revised supplementary information, as also shown below:

Page 13 in the revised manuscript: “We also compared its H₂O₂ production efficiency in pure water and different hole sacrificial agents (Supplementary Fig. 73). It is shown that the yield of H₂O₂ for COF-300-D-F is 1.20 mmol g⁻¹ h⁻¹ in pure water without any organic sacrificial agent. Although the efficiency is lower than that of the system using an organic sacrificial agent, it provides the possibility for its application in a more practical application scenario.”

Supplementary Fig. 73 H₂O₂ generation rates of the photocatalytic reaction of COF-300-D-F with pure water and different sacrificial agents (ethanol, EtOH; methanol, MeOH; isopropanol, IPA; benzyl alcohol, BnOH).

Your point of view is correct. It is the ultimate challenge to realize H₂O₂ production without using sacrificial reagents. The above experiments confirm that the yield of H₂O₂ from COF-300-D-F remains at 1.20 mmol g⁻¹ h⁻¹ in pure water, without the use of any organic sacrificial agent. This proves that COF-300-D-F can directly scavenge photogenerated holes with water molecules through a water oxidation reaction. Although the efficiency is lower than that of the system using an organic sacrificial agent, it provides the possibility for its application in a more practical scenario. The transition to a sacrificial agent-free system requires a material that can effectively drive both the oxygen reduction

reaction and the water oxidation reaction simultaneously. The logical next step we are pursuing now is to integrate a cocatalyst or construct a heterojunction within the backbone of COF, thereby skillfully managing the oxidation semi-reaction and moving towards a completely sustainable process.

Thank you very much for your constructive comments. We believe that this has effectively improved the quality of our work.

Minor issues

(5) Is 20% the upper limit for TAPA that can be incorporated into COF-300-D?

Response to comment:

Thank you for this important question regarding the compositional limits of our COF-300-D series. Your question is very important in helping us improve the quality of our work. According to your suggestion, we synthesized a series of COF-300-D samples with a higher TAPA doping amount under the same conditions and characterized them by PXRD. Results indicate that 20% represents the upper limit for the successful incorporation of TAPA into the COF-300 lattice while maintaining long-range crystalline order. This conclusion is robustly supported by the PXRD data of samples with TAPA doping levels exceeding 20%. The COF-300-D-20% sample retains the characteristic diffraction patterns of the dia-topological network, albeit with some peak broadening, indicating the onset of structural strain. However, when the doping amount of TAPA fraction is increased to 25%, the intensity of the diffraction peaks decreases dramatically, indicating a severe loss of crystallinity. Finally, the PXRD pattern of COF-300-D-35% confirms the complete loss of long-range order and the formation of an amorphous polymer. Determining the upper limit of doping is a valuable discovery in itself, as it defines the practical design space for future functionalization of these COFs. The corresponding details can be found in the revised manuscript and revised supplementary information, as also shown below:

Page 7 in the revised manuscript: “Once the doping amount reaches 25% or more, it may lead to the loss of long-range ordered structure. (Supplementary Fig. 38).”

Supplementary Fig. 38 Experimentally obtained PXRD patterns of COFs.

(6) Details of Pawley refinements should be described in the Supplementary Methods.

Response to comment:

As you suggested, we have added the specific parameters of Pawley refinements to the supplementary information of the paper, as follows:

Supplementary Table 1. Model structures for COF-300, COF-300-D-15%, and COF-300-D-F were generated in BIOVIA Materials Studio 8.0. Simulation of PXRD patterns and Pawley Refinement were performed using the Reflex module.

	Space group	a	c	R_{wp}	R_p
COF-300	$I4_1/a$	27.2354	7.5116	2.68%	1.84%
COF-300-5%	$I4_1/a$	27.1423	7.5418	1.75%	1.26%
COF-300-10%	$I4_1/a$	27.1516	7.5475	1.59%	1.15%
COF-300-15%	$I4_1/a$	27.1179	7.5596	1.68%	1.24%
COF-300-20%	$I4_1/a$	27.1001	7.5577	1.39%	1.01%

We believe that these supplementary details ensure the clarity and repeatability of the intensive revision process, and thank you again for helping us improve the paper.

(7) The space between 0 and 5 in Supplementary Fig. 40 measurement wavelength ($\lambda = 1.54056 \text{ \AA}$) is unnecessary.

Response to comment:

Thank you for your valuable comments. We have deleted the space between 0 and 5 in Supplementary Fig. 40 measurement wavelength ($\lambda = 1.54056 \text{ \AA}$). The corresponding details can be found in the revised supplementary information, as also shown below:

Supplementary Fig. 44 (a, c, e, g, i, k) 2D SAXS image of COFs ($\lambda = 1.54056 \text{ \AA}$). (b, d, f, h, j, l) Experimentally obtained SAXS results (red dot), Pawley refined patterns (black curve), their differences (green curve), and Bragg position (blue bars) of COFs.

(8) The author should add the redox potential for the oxidation of benzyl alcohol to benzyl aldehyde to Fig. 3b.

Response to comment:

Thank you for your valuable comments. We have added the redox potential for the oxidation of benzyl alcohol to benzyl aldehyde to Fig. 3b. The corresponding details can be found in the revised manuscript, as also shown below:

Fig. 3 Photophysical and electrochemical measurements. b, Energy band values of COF-300, COF-300-D-15%, and COF-300-D-F.

Page 13 in the revised manuscript: “In addition, the valence band levels (VB) of COF-300-D-F, COF-300-D-15%, and COF-300 are +1.61, +1.68, and +2.23 V (versus NHE), respectively, which are positive enough to allow the oxidation reaction of benzyl alcohol (BnOH) to benzaldehyde (BD) (Fig. 3b) (*Nat. Commun.* **2024**, 15, 7783; *EES Catal.* **2023**, 1, 552-561).”

(9) Fig. 3d discusses the luminescence intensity of each compound, but is there any information on the photoluminescence quantum yield?

Response to comment:

We sincerely thank the reviewer for this insightful suggestion. As a direct response to your question, we have now measured the absolute photoluminescence quantum yield (PLQY) for our COFs. The results have been incorporated into the revised manuscript and provide crucial quantitative support for our findings. To align all optical characterizations with our photocatalytic testing conditions ($\lambda > 420$ nm), we have re-measured all fluorescence data using a unified excitation wavelength of 480 nm.

The revised Supplementary Fig. 64 now shows the PL spectra under 480 nm excitation. We have also updated the caption and main text to clearly state the excitation wavelength. However, the change in fluorescence intensity we have newly obtained differs from what we originally reported, which was collected at 385 nm. Therefore, we changed the instrument model and made repeated tests at multiple wavelengths. The experimental results confirmed that the COF-300-D-F modified by our strategy exhibited stronger fluorescence. Therefore, at your suggestion, we have carried out a photoluminescence quantum yield (PLQY) test, fluorescence lifetime (τ_{ave}) decay spectra test, and then we calculated the radiative recombination rate (k_r) and nonradiative recombination rate (k_{nr}) according to the formulas $PLQY = \frac{k_r}{k_r + k_{nr}}$ and $\tau_{ave} = \frac{1}{k_r + k_{nr}}$ (*J. Am. Chem. Soc.* **2020**, 142, 16001; *ACS Catal.* **2025**, 15, 15982) (Fig. 3d and 3e). The results showed that PLQY increased from 0.13% for COF-300 to 0.32% for COF-300-D-F, while the fluorescence lifetime increase from 1.82 ns for COF-300 to 3.04 ns for COF-300-D-F. The decrease in k_{nr} and the increase in k_r indicate that, compared with COF-300, the nonradiative recombination of COF-300-D-F is inhibited. Prolonged fluorescence lifetime results in more effective charge separation overall. This is to say that our strategy reduces the overall charge recombination mainly through inhibiting nonradiative recombination, thus improving the apparent quantum yield of the photocatalytic reaction.

We believe this new quantitative data significantly strengthens our discussion on the photophysical properties of the materials and provides a clear mechanism behind the observed fluorescence changes. We are grateful for the reviewer's comment, which has helped us improve the

rigor of our work. The corresponding details can be found in the revised manuscript and revised supplementary information, as also shown below:

Page 10 in the revised manuscript: “In the steady-state photoluminescence spectrum collected under the excitation wavelength of 480 nm (PL, Supplementary Figs. 63-64), it can be observed that, compared to COF-300, the fluorescence intensity gradually increases for COF-300-D-F (*ACS Catal.*, **2025**, 15, 15982-15991). We further measured its photoluminescence quantum yield (PLQY) and fluorescence lifetime. As shown in Supplementary Fig. 65, the PLQY of COF-300, COF-300-D-15%, and COF-300-D-F are 0.13%, 0.16%, and 0.32%, respectively. COF-300-D-F has a significantly longer lifetime of the charge-separated state ($\tau_{ave3} = 3.04$ ns) than COF-300 ($\tau_{ave1} = 1.82$ ns) and COF-300-D-15% ($\tau_{ave2} = 1.96$ ns) (Fig. 3d), implying more efficient charge separation. Such an excited state lifetime reflected the overall carrier recombination rate. Through the quantitative analysis of fluorescence kinetic parameters, the carrier nonradiative recombination rate (k_{nr}) and radiative recombination rate (k_r) are further calculated according to the formulas $PLQY = \frac{k_r}{k_r + k_{nr}}$ and $\tau_{ave} = \frac{1}{k_r + k_{nr}}$ (Fig. 3e) (*J. Am. Chem. Soc.*, **2020**, 142, 16001-16006; *ACS Catal.*, **2025**, 15, 15982-15991). All samples are excited by a 480 nm laser. The decrease in nonradiative recombination and the increase in radiative recombination indicate that, compared with COF-300, the nonradiative recombination of COF-300-D-F is inhibited, resulting in higher visible light activity. This is due to the introduction of the light absorption unit (TAPA) and D-A structure in COF-300-D-F, which can effectively improve the overall photocatalytic efficiency.”

Supplementary Fig. 64 PL spectra of these three COFs at the excitation wavelength of 480 nm.

Supplementary Fig. 65 PLQY collected under 480 nm laser.

Fig. 3 Photophysical and electrochemical measurements. **a**, UV-vis DRS of COF-300, COF-300-D-15%, and COF-300-D-F. **b**, Energy band values of COF-300, COF-300-D-15%, and COF-300-D-F. **c**, Transient photocurrents of COFs under $\lambda > 420$ nm. **d**, Fluorescence lifetime decay spectra of COF-300 (τ_{ave1}), COF-300-D-15% (τ_{ave2}), and COF-300-D-F (τ_{ave3}) at the excitation wavelength of 480 nm and their respective maximum emission wavelengths (600 nm, 650 nm, and 650 nm, respectively). **e**, k_r and k_{nr} obtained by calculation. **f**, Adsorption energy of O_2 on three COFs. **g-i**, Calculation of LUMO/HOMO for the structural segments of COF-300 (**g**), COF-300-D-15% (**h**), and COF-300-D-F (**i**).

Reviewer #2 (Remarks to the Author):

“The manuscript describes a strategy to enhance the photocatalytic activity of 3D covalent organic frameworks (COFs) by partially replacing tetrahedral nodes in COF-300 with planar tri-connected ligands (TAPA) and replacing linear 2-c linkers with electron-withdrawing or -donating groups. This approach tailors light absorption, charge separation, and overall catalytic efficiency. The authors demonstrate very high rates of H₂O₂ production, excellent stability over extended operation, and good performance in benzylamine coupling reactions. The work is well presented and the experiments are solid. The catalytic results are particularly impressive, showing clear advantages over comparable systems in the literature. The characterization is thorough and supports the conclusions drawn. I believe the manuscript is suitable for Nature Communications, although some points could be clarified to improve the overall message.

My main comment concerns how the strategy is framed. The term knot-connectivity reduction strategy gives the impression of precise control over node connectivity. However, what the study really shows is that using mixed ligands with varying connectivity introduces defects in a controlled way up to a point, but with increasing disorder as the proportion of tri-connected ligands rises (PXRD and SAXS). The authors could describe this in a more balanced way, highlighting both the benefits and the structural limits of the approach:”

I recommend minor revision. The study is of high quality, and the suggested clarifications would help sharpen the main message. The manuscript is generally clear and well written, though a few sections could benefit from slightly more concise language.

Response to the general comment:

We sincerely thank you for your positive assessment of our work and for this extremely insightful and constructive comment. We fully agree with your perspective that the term “knot-connectivity reduction strategy” might overemphasize precision and could be more accurately described to reflect the nature of our approach and its inherent structural trade-offs.

In direct response to your suggestion, we have revised the manuscript throughout to reframe our strategy as a “defect engineering” approach. This terminology more accurately conveys the concept of

intentionally incorporating planar tri-connected ligands into a tetrahedral framework to create beneficial defects and tune electronic properties, while also acknowledging the potential for introduced disorder.

Specifically, we have made the following changes:

First of all, in our work, the term “knot-connectivity reduction strategy” has been replaced with “defect engineering”. Second, in the discussion of structural evolution, we have now clearly emphasized the balance between benefits and the structural limitations of the approach. Last but not least, as suggested, we have included a short discussion on the limits of this approach in the concluding part, noting that the framework's crystallinity can only tolerate the incorporation of the lower-connectivity ligand up to a certain percentage (20% in our system), beyond which the structural integrity is compromised.

We are grateful for this valuable comment, which has helped us to present our work with greater clarity and scientific rigor. We believe the revised manuscript now more accurately and effectively communicates the significance and scope of our findings.

(1) I suggest the authors reconsider the way they describe their strategy. Rather than “knot-connectivity reduction,” it might be clearer to present it as a mixed-ligand or defect engineering approach that allows introduction of planar units and tuning of electronic properties, while acknowledging the gradual loss of order at higher TAPA content. Similar strategies are followed in metal-organic frameworks and are often referred to as ligand-truncation.

Response to comment:

Thank you very much for your valuable and insightful suggestions. We totally agree with you that using a more accurate description can make our strategy clearer and easier for readers to understand. According to your suggestion, we have completely revised the manuscript. The main changes are as follows:

Revised Terminology: We have replaced the phrase “knot-connectivity reduction strategy” throughout the manuscript with the more standard and descriptive term “defect engineering”.

Acknowledgment of Structural Order: We observed that the full width at half maximum (FWHM) of the main diffraction peaks in the COF-300-D series increased with the increase in the doping amount of TAPA. This indicates that the structural order of the COF-300-D series is weakened at higher defect concentrations, a common phenomenon in defective engineering crystal materials. As suggested, we have included a brief discussion on the limitations of this approach in the concluding section, noting that the framework's crystallinity can only tolerate the incorporation of the lower-connectivity ligand up to a certain percentage (20% in our system), beyond which the structural integrity is compromised. The details can be found in the revised manuscript, as follows:

Page 7 in the revised manuscript: “The full width at half maximum (FWHM) values reflected the coherence of the structure across the entire crystal samples. It is worth noting that with the increase in TAPA content, the FWHM increases gradually (Supplementary Fig. 46), indicating that structural disorder accompanies the increment, in which the introduction of functional nodes and crystal integrity are balanced. The relative narrow FWHM of 200, 220, and 400 reflections at low q range further confirmed the preservation of the integrity of the COF local structure as TAPA units were introduced.”

Supplementary Fig. 46 Full width at half maximum (FWHM) of the main diffraction peaks in the SAXS patterns.

We believe these revisions have significantly improved the clarity and precision of our manuscript. We are grateful to the reviewer for this valuable feedback.

(2) It would strengthen the paper to include a short discussion on the limits of this approach. For example, what is the maximum amount of low-connectivity ligand the structure can tolerate before crystallinity is lost or the dia net is not assembled? This would help readers better understand how the method could be applied to other COF systems.

Response to comment:

We are very grateful to you for this excellent suggestion. We agree that discussing the limitations of our defect engineering strategy is crucial to understanding its applicability and scope. In response, we discuss in detail the maximum doping amount of low-linking ligands that can be tolerated by the defect engineering strategy and provide necessary exposition in the discussion section.

Specifically, we synthesized a series of COF-300-D samples with a higher TAPA doping amount under the same conditions and characterized them by PXRD. Results indicate that 20% represents the upper limit for the successful incorporation of TAPA into the COF-300 lattice while maintaining long-range crystalline order. This conclusion is robustly supported by the PXRD data of samples with TAPA doping levels exceeding 20%. The COF-300-D-20% sample retains the characteristic diffraction patterns of the dia-topological network, albeit with some peak broadening, indicating the onset of structural strain. However, when the doping amount of TAPA fraction is increased to 25%, the intensity of the diffraction peaks decreases dramatically, indicating a severe loss of crystallinity. Finally, the PXRD pattern of COF-300-D-35% confirms the complete loss of long-range order and the formation of an amorphous polymer. Determining the upper limit of doping is a valuable discovery in itself, as it defines the practical design space for future functionalization of these COFs. The corresponding details can be found in the revised manuscript and revised supplementary information, as also shown below:

Page 20 in the revised manuscript: “However, defect introduction exhibits a critical threshold beyond which accumulated lattice strain compromises crystallinity. This strategy can be extended to other 3D COF platforms, where identifying similar compositional limits will be essential. Moving forward, designing frameworks with higher intrinsic tolerance for structural diversity or employing geometrically compatible linkers offers a promising path to multifunctional COFs that maintain

structural integrity while integrating advanced functionalities.”

Page 7 in the revised manuscript: “Once the doping amount reaches 25% or more, it may lead to the loss of long-range ordered structure (Supplementary Fig. 38).”

Supplementary Fig. 38 Experimentally obtained PXRD patterns of COFs.

Reviewer #3 (Remarks to the Author):

“In this study, the authors introduce a knot-connectivity reduction strategy to construct 3D COFs for hydrogen peroxide photocatalysis and photocatalytic benzylamine coupling reactions. I highly support its publication in Nature Communications after sufficient revisions with the following considerations:”

Response to the general comment:

We sincerely thank the reviewer for the thorough assessment of our manuscript and for the highly supportive and constructive comments. We are greatly encouraged by the reviewer’s positive assessment that our work on the knot-connectivity reduction strategy for 3D COFs is suitable for Nature Communications after revisions. We have carefully considered all the comments and have performed extensive additional experiments, characterizations, and textual revisions to address every point raised. In our view, these revisions greatly enhanced the clarity, depth, and overall scientific rigor of the manuscript. Below, we have provided a point-by-point response to each comment. All changes in the manuscript have been highlighted for the reviewer's convenience. We believe the revised manuscript now fully meets the high standards of the journal and hope that our responses are satisfactory.

(1) The authors did not determine the actual structure of the series of 3D COFs, especially without considering the impact of the node replacement strategy on the interpenetration structure of dia topology network. Therefore, with the increase of defect concentration in COFs, does the full width at half maximum (FWHM) of diffraction peaks in PXRD spectra show significant changes? Is the distribution of these defects uniform? What about the pore size distribution in the COF-300-D series, and is it allowed for benzylamine to enter the pores and contact with catalytic sites?

Response to comment:

Thank you for your profound questions about the structural integrity, distribution uniformity, and porosity of our COF-300-D series COFs. These points are very important for understanding the structure-performance relationship of defective engineering materials.

a) On the change of PXRD FWHM and defect distribution:

You correctly predicted that introducing defects may lead to the peak broadening of PXRD. In addition, our experimental data also meet this expectation and provide nuanced insights. We observed that the FWHM of the main diffraction peaks in the COF-300-D series increased with the increase in the doping amount of TAPA. This indicates that the structural order of the COF-300-D series is weakened at higher defect concentrations, a common phenomenon in defective engineering crystal materials. In addition, the relative narrow FWHM of 200, 220, and 400 reflections at low q range further confirmed the preservation of the integrity of the COF local structure as TAPA units were introduced. Interestingly, we found that COF-300-D-15% showed slightly lower FWHM for several specific reflections. It is speculated that, with this specific composition, TAPA units were doped into the framework in a coherent manner, thus minimizing damage to the crystal lattice.

As both occupational and displacement disorders occur during crystal growth, it will be hardly to obtain the atomic distribution of these TAPA molecular across the entire crystal by diffraction methods. Considering the fluorescence nature of TAPA linkers, we applied laser scanning confocal microscopy (LSCM) to visualize the distribution of TAPA linkers throughout the single crystal of COF-300-D-15%. The precise distribution of TAPA across COF-300-D-15% single-crystal was mapped layer by layer at different focal depths with about 0.2 μm intervals to reconstruct 3D tomography (Fig. 2h). Red fluorescence was observed in all layers under excitation light of 405 nm, indicating the homogeneous and random distribution of TAPA across the whole crystal, while COF-300 exhibits no fluorescence under the same condition (Supplementary Figs. 49-50).

Page 7 in the revised manuscript: “The full width at half maximum (FWHM) values reflected the coherence of the structure across the entire crystal samples. It is worth noting that with the increase in TAPA content, the FWHM increases gradually (Supplementary Fig. 46), indicating that structural disorder accompanies the increment, in which the introduction of functional nodes and crystal integrity are balanced. The relative narrow FWHM of 200, 220, and 400 reflections at low q range further confirmed the preservation of the integrity of the COF local structure as TAPA units were introduced.”

Supplementary Fig. 46 Full width at half maximum (FWHM) of the main diffraction peaks in the SAXS patterns.

Page 7 in the revised manuscript: “The atomic distribution of TAPA in COF-300-D was hard to determine using X-ray or electron diffraction techniques due to both occupational and displacement disorders. Considering the fluorescence nature of TAPA linkers, we applied laser scanning confocal microscopy (LSCM) to visualize the distribution of TAPA linkers throughout the single crystal of COF-300-D-15%. The precise distribution of TAPA across COF-300-D-15% single-crystal was mapped layer by layer at different focal depths with about 0.2 μm intervals to reconstruct 3D tomography (Fig. 2h). Red fluorescence was observed in all layers under excitation light of 405 nm, indicating the homogeneous and random distribution of TAPA across the whole crystal, while COF-300 exhibits no fluorescence under the same condition (Supplementary Figs. 49-50).”

b) On pore size distribution and benzylamine accessibility:

Thank you very much for raising this profound question. It is very important that you pay attention to the accessibility of reactants. We are here to explain the question in detail by combining the pore size data and flexibility characteristics of the materials.

The analysis of pore size distribution by the nonlocal density functional theory model shows that the average pore size of COF-300, COF-300-D-15%, and COF-300-D-F is about 1.08 nm. It is proven that the basic pore structure of dia topology is preserved after TAPA doping. The corresponding details can be found in the revised manuscript and revised supplementary information, as also shown below:

Page 8 in the revised manuscript: “The analysis of pore size distribution by the nonlocal density functional theory model shows that the average pore size of COF-300, COF-300-D-15%, and COF-300-D-F is about 1.08 nm (Supplementary Fig. 53).”

Supplementary Fig. 53 Pore size distributions of COFs.

We believe that benzylamine can effectively enter the pores and contact with catalytic sites for the following reasons. Comparison of thermodynamic diameters: The kinetic diameter of benzylamine is estimated to be approximately 7.144 Å, which is equivalent to 0.71 nm. For the COF-300-D series with a pore size of 1.08 nm, benzylamine with a pore size of 0.71 nm can diffuse freely into its pores, as shown in Response Fig. 1. Furthermore, unlike many porous crystal materials, COF-300 and its derivatives are not completely rigid structures, but have inherent flexible dynamic response characteristics (*J. Am. Chem. Soc.* **2019**, 141, 3298–3303; *J. Am. Chem. Soc.* **2024**, 146, 1035–1041). This means that in a practical environment (especially in a liquid reaction medium), the pores of the material can expand or adjust their conformation to some extent. When encountering molecules with a similar size, the skeleton of COF can adapt to the entry and exit of molecules through slight deformation. Therefore, the effective pore diameter of COF-300-D series in solution will be slightly larger than its theoretical pore diameter measured by gas adsorption in the dry state. This dynamic characteristic ensures that 0.71 nm benzylamine molecules can smoothly enter the internal space

through the original hole window of 1.08 nm. In addition, the excellent catalytic efficiency of our photocatalytic benzylamine coupling reaction also decisively confirmed this accessibility.

Response Fig. 1 Schematic diagram of benzylamine with optimized structure and COF-300-D series.

(2) *The authors should consider the possibility of BDA monomers being connected to TAPA monomers at both ends in the COF-300-D series, rather than the idealized structure of one end TAPA and the other end TAM, especially when the concentration of TAPA increases.*

Response to comment:

We sincerely thank you for putting forward this excellent and key point of view. We fully agree that in the mixed ligand strategy, it is possible that the BDA monomer will be linked to two TAPA monomers at higher TAPA concentration. Our initial description of TAM-BDA-TAPA is a simplified model to illustrate this concept. In fact, connectivity is more complicated. This is an important consideration that we should solve, and we are glad to have the opportunity to improve our structural model. We thank you again for this insightful comment, which enables us to describe the structure-

performance relationship in defective engineering COF more accurately. The corresponding details can be found in the revised supplementary information, as also shown below:

Supplementary Fig. 3 a, Schematic diagram for the synthesis of the series of COFs with different contents of TAPA (COF-300-D) ($D = 5\%$, 10% , 15% , 20%). **b**, Schematic diagram of possible conformations of COF-300-D series. The doping of mixed ligands follows a statistical distribution. With the increase in TAPA content, the probability of both ends of the BDA connector being connected to the plane TAPA nodes increases significantly, which may lead to a decrease in the long-range order of the COF-300-D series with a high doping ratio.

(3) The ^1H and ^{13}C NMR did not show significant differences in the Supplementary Figures. How to calculate Supplementary Fig. 35 based on ^1H NMR? More details need to be provided.

Response to comment:

We sincerely appreciate your insightful and constructive comments, which help a lot to enhance the quality of our manuscript. In the digestion ^{13}C NMR of COF-300 and COF-300-D series (Supplementary Fig. 10 to 14), the characteristic peaks of TAPA (labeled 10 and 11 at ~ 125 ppm) are exclusively observed in COF-300-D, with no corresponding signals detected in COF-300. In the digestion ^1H NMR (Supplementary Figs. 24-28), the characteristic peaks of TAPA (labeled 5 and 6) are overlapped with those from H_3O^+ and TAM, making interpretation complicated. However, the molar ratio of TAM: BDA in COF-300 is fixed at 1:2, while the molar ratio of (TAM+TAPA):BDA in COF-300-D is also approximated to 1:2. This enables the BDA can be treated as an internal standard. Noting that the reduced total integration for peaks 3, 4, and 6 in COF-300-D (vs. peaks 3 and 4 in COF-300) originates from TAPA doping, as TAPA contains fewer hydrogen atoms than TAM. The number of protons represented by peaks 3, 4, and 6 can be expressed as $n(1-f_{\text{TAPA}}) + m \times f_{\text{TAPA}}$. Thus, we could build the formula:

$$k \times \frac{I(\text{H}_{3,4,6})}{I(\text{H}_1)} = n(1 - f_{\text{TAPA}}) + m \times f_{\text{TAPA}}$$

Where **k** is the number of protons that contribute to $I(\text{H}_1)$ from the BDA unit corresponding to one TAM unit in COF-300, here is a constant 4. **I(H_{3,4,6})** is the integrated area of the overlapping peaks (H-3, H-4 of TAM and H-6 of TAPA), **I(H₁)** is the integrated area of the peak (H-1), **n** is the number of protons from one TAM unit that contribute to $I(\text{H}_{3,4,6})$, and here is a constant 16. **m** is the number of protons from one TAPA unit that contribute to $I(\text{H}_{3,4,6})$; here, it is a constant of 6.

Your careful examination is greatly appreciated, and more details and further corrections are provided in the revised supplementary information, as also illustrated below:

Supplementary Fig. 10 Digestion ^{13}C NMR spectrum of the activated COF-300.

Supplementary Fig. 11 Digestion ^{13}C NMR spectrum of the activated COF-300-D-5%.

Supplementary Fig. 12 Digestion ¹³C NMR spectrum of the activated COF-300-D-10%.

Supplementary Fig. 13 Digestion ¹³C NMR spectrum of the activated COF-300-D-15%.

Supplementary Fig. 14 Digestion ^{13}C NMR spectrum of the activated COF-300-D-20%.

Page S65 in the revised supplementary information: “The actual molar fraction of TAPA (f_{TAPA}) in the COF-300-D series was determined via ^1H NMR analysis of the digested samples (Supplementary Fig. 24-28). The well-resolved signal from a designated proton of the BDA linker (H_1 , $\delta = 10.17$ ppm) was used as an internal standard. This is justified because the molar ratio of (TAM + TAPA) to BDA is fixed at 1:2 in the COF structure, ensuring a constant reference point.

The calculation was based on the integrated area of the overlapping signals $I(\text{H}_{3,4,6})$, which includes H-3 and H-4 of TAM and H-6 of TAPA ($\delta 7.37\text{-}7.44$ ppm). The mole fraction f_{TAPA} was calculated using the following equation derived from proton count conservation:

$$f_{TAPA} = \frac{\left[k \times \frac{I(\text{H}_{3,4,6})}{I(\text{H}_1)} \right] - n}{m - n} \times 100\% \quad (1)$$

Where k is the number of protons that contribute to $I(\text{H}_1)$ from the BDA unit corresponding to one TAM unit in COF-300, here is a constant 4. $I(\text{H}_{3,4,6})$ is the integrated area of the overlapping peaks (H-3, H-4 of TAM and H-6 of TAPA), $I(\text{H}_1)$ is the integrated area of the peak (H-1), n is the number of protons from one TAM unit that contribute to $I(\text{H}_{3,4,6})$, and here is a constant 16. m is the number of protons from one TAPA unit that contribute to $I(\text{H}_{3,4,6})$; here, it is a constant of 6.

The equation is derived as follows: The ratio $I(H_{3,4,6})/I(H_1)$ is proportional to the number of protons in the overlap region per BDA unit. Therefore:

$$k \times \frac{I(H_{3,4,6})}{I(H_1)} = n(1 - f_{TAPA}) + m \times f_{TAPA} \quad (2)''$$

(4) *Is this still a good strategy for the quantitative ratio of COF-320 series? And how about the photocatalytic performance of COF-320 series in producing hydrogen peroxide?*

Response to comment:

We sincerely thank the reviewer for these insightful questions regarding the generality of our mixed-ligand strategy to the COF-320 series. The reviewer's comments prompted us to conduct further experiments to fully address these points, which have significantly strengthened our conclusions. Following the reviewer's suggestion, we have synthesized the COF-320-D-F sample (Supplementary Fig. 42) and evaluated its photocatalytic performance for H₂O₂ production. The results showed that the photocatalytic efficiency was drastic improved after the introduction of TAPA and the electronic modulator (BPDA-F) (Supplementary Fig. 74). These new experimental results, which have been added to the revised manuscript and revised supplementary information, are highly encouraging:

Page S69 in the revised supplementary information: “**Synthesis of COF-320-D-F:** COF-320-D-F was synthesized by a one-step solvothermal method and modified on the basis of COF-320-D-15%. A Pyrex tube measuring 10 × 8 mm (o.d × i.d) was charged with TAM (16.4 mg, 0.043 mmol), TAPA (2.3 mg, 0.008 mmol), 4,4'-biphenyldialdehyde (BPDA) (10.5 mg, 0.05 mmol), [1,1'-Biphenyl]-4,4'-dicarboxaldehyde, 2,2',3,3',5,5',6,6'-octafluoro- (BPDA-F) (17.7 mg, 0.05 mmol), 1,4-dioxane (0.25 mL), and pyridine (0.15 mL), aniline (0.06 mL) and 0.2 mL of 6 M aqueous acetic acid water. The subsequent operations are the same as those of COF-320-D-15%. The synthesis diagram is shown in Supplementary Fig. 42.”

Supplementary Fig. 42 Schematic diagram of the synthesis of COF-320-D-F. To validate the universal applicability of this defect engineering, COF-320-D-15% and COF-320-D-F were synthesized using an identical protocol.

Page 14 in the revised manuscript: “Furthermore, COF-320-D-F exhibited an excellent rate of $4.24 \text{ mmol g}^{-1} \text{ h}^{-1}$, which was 7.6 and 2.6 times that of COF-320 ($0.56 \text{ mmol g}^{-1} \text{ h}^{-1}$) and COF-320-D-15% ($1.63 \text{ mmol g}^{-1} \text{ h}^{-1}$), respectively (Supplementary Fig. 74). This confirms the potential of the defect engineering strategy in the functionalization of 3D COF.”

Supplementary Fig. 43 Experimental PXRD patterns of COF-320, COF-320-D-15%, and COF-320-D-F. Structural characterization revealed well-preserved crystallinity with minimal structural alterations.

Supplementary Fig. 74 Photocatalytic activity of COF-320, COF-320-D-15%, and COF-320-D-F for H₂O₂ generation.

In summary, we are grateful to the reviewer for prompting us to demonstrate the broader applicability of our strategy. The new data on the COF-320 series not only address the reviewer's questions but also elevate the significance of our work by showing that the mixed-ligand strategy is a generalizable design principle for enhancing the photocatalytic activity of 3D COFs with dia topology.

(5) *Why is the fluorescence emission peak of COF-300 in the Fig. 3d around 520 nm, while in Supplementary Fig. 60 the fluorescence emission peak of the three-dimensional fluorescence spectrum is around 600 nm? Does the lack of normalization of fluorescence intensity in Fig. 3d indicate a difference in the relative fluorescence intensity of these COFs? What are the excitation and emission wavelengths corresponding to the fluorescence lifetime in Fig. 3e?*

Response to comment:

We sincerely thank the reviewers for posing these insightful questions, which prompted us to conduct more rigorous and consistent spectral analysis, thereby significantly improving our data quality and understanding of the mechanism.

Firstly, the obvious difference in the fluorescence emission peak is caused by the different excitation wavelengths used in these two different measurements. Our original Fig. 3d used an excitation wavelength of 385 nm, which was selected based on common practices in the initial literature. We have further supplemented the three-dimensional fluorescence spectrum with a wider range of excitation wavelengths to answer your question, as shown in Response Fig. 2. However, the excitation wavelength of the fluorescence emission center of COFs is all in the range of 480 ~ 490 nm (Supplementary Fig. 63), and the photocatalytic activity is tested under visible light ($\lambda > 420$ nm). To resolve this and align all optical characterizations with our photocatalytic testing conditions, we have re-measured all fluorescence data using a unified excitation wavelength of 480 nm.

Response Fig. 2 Three-dimensional fluorescence spectrum of COF-300 (a), COF-300-D-15% (b), and COF-300-D-F (c).

Supplementary Fig. 63 Three-dimensional fluorescence spectrum of COF-300 (a), COF-300-D-15% (b), and COF-300-D-F (c).

The revised Supplementary Fig. 64 now shows the PL spectra under 480 nm excitation. The maximum emission peak for COF-300 is indeed observed at approximately 600 nm, which is fully consistent with the peak position in the 3D fluorescence spectrum (Supplementary Fig. 63). We have also updated the caption and the main text to clearly state the excitation wavelength. In the PL spectrum collected under an excitation wavelength of 480 nm, it can be observed that, compared to COF-300, the fluorescence intensity of COF-300-D-F increases gradually. Furthermore, we employ a more systematic photoluminescence quantum yield (PLQY) test, fluorescence lifetime (τ_{ave}) decay spectra test, and quantitative analysis of fluorescence kinetic parameters to characterize the excited state behavior of COFs more systematically, thereby exploring the underlying mechanism. We calculate the radiative recombination rate (k_r) and nonradiative recombination rate (k_{nr}) according to the formulas $\text{PLQY} = \frac{k_r}{k_r + k_{nr}}$ and $\tau_{\text{ave}} = \frac{1}{k_r + k_{nr}}$ (*J. Am. Chem. Soc.* **2020**, 142, 16001; *ACS Catal.* **2025**, 15, 15982) (Fig. 3d and 3e). The results showed that PLQY increased from 0.13% for COF-300 to 0.32% for COF-300-D-F, while the fluorescence lifetime increase from 1.82 ns for COF-300 to 3.04 ns for COF-300-D-F. The decrease in k_{nr} and the increase in k_r indicate that, compared with COF-300, the

nonradiative recombination of COF-300-D-F is inhibited. Prolonged fluorescence lifetime results in more effective charge separation overall. This is to say that our strategy reduces the overall charge recombination mainly through inhibiting nonradiative recombination, thus improving the apparent quantum yield of the photocatalytic reaction. The corresponding details can be found in the revised manuscript and revised supplementary information, as also shown below:

Supplementary Fig. 64 PL spectra of these three COFs at the excitation wavelength of 480 nm.

Supplementary Fig. 65 PLQY excited at 480 nm laser.

Page 10 in the revised manuscript: “In the steady-state photoluminescence spectrum collected under the excitation wavelength of 480 nm (PL, Supplementary Figs. 63-64), it can be observed that, compared to COF-300, the fluorescence intensity gradually increases for COF-300-D-F (*ACS Catal.*, **2025**, 15, 15982-15991). We further measured its photoluminescence quantum yield (PLQY) and fluorescence lifetime. As shown in Supplementary Fig. 65, the PLQY of COF-300, COF-300-D-15%, and COF-300-D-F are 0.13%, 0.16%, and 0.32%, respectively. COF-300-D-F has a significantly longer lifetime of the charge-separated state ($\tau_{ave3} = 3.04$ ns) than COF-300 ($\tau_{ave1} = 1.82$ ns) and COF-300-D-15% ($\tau_{ave2} = 1.96$ ns) (Fig. 3d), implying more efficient charge separation. Such an excited state lifetime reflected the overall carrier recombination rate. Through the quantitative analysis of fluorescence kinetic parameters, the carrier nonradiative recombination rate (k_{nr}) and radiative recombination rate (k_r) are further calculated according to the formulas $PLQY = \frac{k_r}{k_r + k_{nr}}$ and $\tau_{ave} = \frac{1}{k_r + k_{nr}}$ (Fig. 3e) (*J. Am. Chem. Soc.*, **2020**, 142, 16001-16006; *ACS Catal.*, **2025**, 15, 15982-15991). All samples are excited by a 480 nm laser. The decrease in nonradiative recombination and the increase in radiative recombination indicate that, compared with COF-300, the nonradiative recombination of COF-300-D-F is inhibited, resulting in higher visible light activity. This is due to the introduction of the light absorption unit (TAPA) and D-A structure in COF-300-D-F, which can effectively improve the overall photocatalytic efficiency.”

Fig. 3 Photophysical and electrochemical measurements. **a**, UV-vis DRS of COF-300, COF-300-D-15%, and COF-300-D-F. **b**, Energy band values of COF-300, COF-300-D-15%, and COF-300-D-F. **c**, Transient photocurrents of COFs under $\lambda > 420$ nm. **d**, Fluorescence lifetime decay spectra of COF-300 (τ_{ave1}), COF-300-D-15% (τ_{ave2}), and COF-300-D-F (τ_{ave3}) at the excitation wavelength of 480 nm and their respective maximum emission wavelengths (600 nm, 650 nm, and 650 nm, respectively). **e**, k_r and k_{nr} obtained by calculation. **f**, Adsorption energy of O_2 on three COFs. **g-i**, Calculation of LUMO/HOMO for the structural segments of COF-300 (**g**), COF-300-D-15% (**h**) and COF-300-D-F (**i**).

We thank the reviewers again for their careful scrutiny. For the fluorescence lifetime decays presented in Fig. 3e of the original submission, the measurements were conducted with an excitation wavelength of 480 nm and emission at 650 nm. Furthermore, to improve data quality and reproducibility, we have repeated the entire fluorescence lifetime measurement series on a different instrument following your review. We have now re-measured the fluorescence lifetimes for all samples using their respective maximum emission wavelengths (600 nm, 650 nm, and 650 nm, respectively). The results from this independent validation, which are now presented in the revised Fig. 3d, consistently reproduce the key finding—a significant elongation of the fluorescence lifetime in the COF-300-D-15% and COF-300-D-F. The revised Fig. 3d and its caption in the manuscript now explicitly state these optimized detection wavelengths. This update provides a more precise representation of the excited-state decay dynamics. We thank the reviewer for prompting us to provide these essential details, which will further solidify the reliability of our conclusions.

In summary, your comments have prompted a comprehensive re-evaluation of our optical data. The revised figures and the discussion in the main text now all consistently refer to data acquired under 480 nm excitation, creating a coherent and scientifically accurate narrative that is directly relevant to the photocatalytic performance. We are grateful that your feedback has led to a substantial enhancement in the quality of our work.

(6) Is there a problem of phase separation or purity at higher doping ratios (over 20%)? Why does the catalytic efficiency decrease when the defect concentration is higher (20%) in Fig. 4a? How does defect concentration affect the mass transfer rate in photocatalytic reactions?

Response to comment:

Thank you very much for your insights and valuable questions. We synthesized a series of COF-300-D samples with varying TAPA doping amounts under identical conditions and characterized them using PXRD. Results indicate that 20% represents the upper limit for the successful incorporation of TAPA into the COF-300 lattice while maintaining long-range crystalline order. The COF-300-D-20% sample retains the characteristic diffraction patterns of the dia-topological network, albeit with some

peak broadening indicating the onset of structural strain. However, when the doping amount of TAPA increased to 25%, the intensity of the diffraction peaks decreased dramatically, indicating a severe loss of crystallinity. Finally, the PXRD pattern of COF-300-D-35% confirms the complete loss of long-range order and the formation of an amorphous polymer. Determining the upper limit of doping is a valuable discovery in itself, as it defines the practical design space for future functionalization of these COFs. The corresponding details can be found in the revised manuscript and revised supplementary information, as also shown below:

Page 7 in the revised manuscript: “Once the doping amount reaches 25% or more, it may lead to the loss of long-range ordered structure. (Supplementary Fig. 38).”

Supplementary Fig. 38 Experimentally obtained PXRD patterns of COFs.

Furthermore, the non-monotonic trend in H_2O_2 production through photocatalysis is indeed an interesting result of our research. We are very pleased to have the opportunity to elaborate on the underlying mechanism in detail. We believe that as the amount of TAPA increases, this phenomenon is caused by the competition between two opposing effects: the introduction of beneficial light-harvesting units and the gradual disruption of the long-range order and crystallinity of the COF framework. The optimal performance at 15% TAPA represents the balance between these two competing factors. At this concentration, the positive effects of enhanced charge separation and the introduction of new active sites maximize the photocatalytic output, while the negative impact of

structural disorder is still minimized. Further increasing the TAPA content would disrupt this balance, as the adverse effects of crystallinity loss and framework disorder outweigh the benefits of introducing active sites, resulting in the observed decrease in activity. This structure-property relationship underscores the importance of precise component control in multivariate COFs for enhancing photocatalytic performance.

To elucidate the influence of defect concentration on mass transport, we constructed a COF-based molecular dynamics (MD) simulation system containing 1475 H₂O molecules and 20 O₂ molecules, confined within a periodic cell of 5.22×5.22×3.77 nm³. To model defective COF frameworks consistent with experimental synthesis, the primitive COF-300 unit cell (*Science*, **2018**, 361, 48 – 52) was first expanded into a 2×2×5 supercell, resulting in the above periodic dimensions. Structural defects were then introduced by randomly substituting a fraction of the tetra(4-aminophenyl)methane (TAM) nodes with tris(4-aminophenyl)amine (TAPA) nodes, mimicking the experimental defect formation process in which TAPA partially replaced TAM during solvothermal condensation. Specifically, 5% and 15% of the TAM units were randomly substituted to construct COF-300-D-5% and COF-300-D-15%, respectively.

As shown in Response Fig. 3, the self-diffusion coefficient of O₂ increases progressively with defect concentration from 0% to 15%, whereas that of H₂O remains essentially unchanged. This trend may be associated with the distinct transport behaviors of the two species within the porous channels. H₂O molecules tend to form a hydrogen-bond-mediated confined water layer near the pore walls. Although this environment imposes spatial restriction, hydrogen bonds can rapidly break and reform, enabling a hopping-like diffusion mechanism (*Science*, **2006**, 311, 832-835). Therefore, water diffusion is likely governed by the intrinsic rearrangement dynamics of the hydrogen-bond network, rendering it relatively insensitive to minor structural perturbations. In contrast, O₂ does not participate in the hydrogen-bond network and diffuses in the form of solvent cages. Its transport is therefore more dependent on the accessible free volume and the tortuosity of the diffusion pathways. Thus, the introduction of defects may increase the accessible pore space and lower pathway tortuosity, potentially facilitating the selective diffusion of O₂.

Response Fig. 3 a, Representative snapshot of the COF-300-D-15% system after MD simulation. The gray region corresponds to the aqueous environment, the red spheres denote O₂ molecules, and the green framework illustrates the COF skeleton (visualized using different rendering styles for clarity). **b**, Self-diffusion coefficients of COFs with different defect concentrations, obtained from the final stage of the NVT production run.

MD simulation details: All simulations were performed using GROMACS 2023.5 (*J. Comput. Chem.* **2005**, 26, 1701). The force-field parameters of the COF framework were generated using Sobtop based on the General Amber force field 2 (GAFF2) (*J. Comput. Chem.* **2004**, 25, 1157), and atomic charges were assigned via the PACMAN-Charge method (*J. Chem. Theory Comput.* **2024**, 20, 5368-5380). Water molecules were described by the SPC/E (extended simple point charge) model (*J. Phys. Chem.* **1987**, 91, 6269), and parameters for O₂ were taken from a reported model (*J. Chem. Theory Comput.* **2021**, 17, 5198-5213). After energy minimization, the system was equilibrated through 200 ps NVT (constant number, volume, and temperature) and 1 ns NPT (constant number, pressure, and temperature) pre-equilibration stages to stabilize the temperature and pressure. Subsequently, a 10 ns NVT production run was carried out at 298 K using the V-rescale thermostat. Long-range electrostatic interactions were treated using particle-mesh Ewald (PME) (*J. Chem. Phys.* **1993**, 98, 10089), and a cutoff of 1.2 nm was applied for van der Waals interactions. Trajectories were saved every 2 ps, and structural and diffusion analyses were performed using the Visual Molecular Dynamics (VMD) software.

(7) What is the photocatalytic efficiency of COF-300-D series for hydrogen peroxide production in pure water and different hole scavengers?

Response to comment:

We are grateful for your important question about the photocatalytic efficiency of different sacrificial agents, which will be very helpful in improving the quality of our manuscript. We tested the yield of H₂O₂ for COF-300-D-F in pure water and different sacrificial agents, and added detailed discussion in the revised manuscript and revised supplementary information, as also shown below:

Page 13-14 in the revised manuscript: “We also compared its H₂O₂ production efficiency in pure water and different hole sacrificial agents (Supplementary Fig. 73). It is shown that the yield of H₂O₂ for COF-300-D-F is 1.20 mmol g⁻¹ h⁻¹ in pure water without any organic sacrificial agent. Although the efficiency is lower than that of the system using an organic sacrificial agent, it provides the possibility for its application in a more practical scenario. Photocatalytic experiments of COF-300-D-F under different sacrificial agents showed that BnOH is the best, which indicates that a two-phase reaction system composed of water and BnOH could strongly facilitate the reaction, because the active center of COFs remains in the BnOH phase, and the generated H₂O₂ quickly diffuses into the water.”

Supplementary Fig. 73 H₂O₂ generation rates of the photocatalytic reaction of COF-300-D-F with pure water and different sacrificial agents (ethanol, EtOH; methanol, MeOH; isopropanol, IPA; benzyl alcohol, BnOH).

(8) *Theoretical calculations should use periodic 3D COFs structures rather than local molecular segments, because it is necessary to take into account the stacking effect and charge transfer between structural units caused by the interweaving of topological structures.*

Response to comment:

We fully agree with the reviewer that the stacking effect and inter-layer charge transfer within 3D COF frameworks are important for understanding their overall electronic characteristics. However, performing periodic density functional theory (DFT) calculations on such large-scale 3D COF systems is computationally prohibitive. The smallest unit cell of our COF already contains 292 atoms, and to model defect concentrations of 5%, 10%, 15%, and 20%, the unit cell must be expanded at least five times, resulting in approximately 1460 atoms. At this scale, periodic DFT calculations become extremely time-consuming and difficult to converge. To balance computational feasibility and physical relevance, we therefore employed representative molecular fragments to approximate the local coordination environment of the active sites. This approach has been widely adopted in COF photocatalysis studies (e.g., *Nat. Commun.* **2023**, 14, 3083; *J. Am. Chem. Soc.* **2025**, 147, 36071-36078; *Angew. Chem. Int. Ed.* **2025**, 64, e202500937). Moreover, the local model can still partially capture the influence of conjugated topology and charge delocalization around the active sites, which are the key regions determining catalytic behavior. Hence, we believe this treatment is both reasonable and representative for describing the essential electronic structure of the catalytic centers.

We thank all referees again for their helpful comments and suggestions, and hope that this significantly revised manuscript is now acceptable for publication in *Nature Communications*.

Best Regards,

Yours Sincerely,

Prof. Dr. Chong Cheng (on behalf of the authors)

Point-by-point response to the detailed comments by reviewers of “*Molecular Engineering of Defective 3D Covalent Organic Frameworks for Enhanced Photocatalytic H₂O₂ Synthesis and Organic Transformation*” with manuscript ID: NCOMMS-25-38777B.

REVIEWER COMMENTS

Reviewer #1 (Remarks to the Author):

“Thank you for responding to my comments. The authors' responses are partially incomplete; therefore, I could reconsider publishing this manuscript if they fully address the comments below.”

Response to the general comment:

Thank you very much for your careful re-evaluation of our manuscript and for your constructive comments. We sincerely appreciate the time and effort you have devoted to assessing our work.

We have now thoroughly revised the manuscript to fully address all of the comments raised in your latest review. In this revised version, we have provided clearer explanations and added the requested analyses and discussions. We believe that these revisions significantly improve the clarity, rigor, and overall quality of the manuscript.

All changes have been carefully indicated in the revised manuscript, and detailed point-by-point responses are provided below for your convenience. We hope that the revisions satisfactorily resolve your concerns and demonstrate the scientific merit and reliability of our work.

About Question (1),

The authors mentioned that the non-monotonic trend in H₂O₂ production is related to the competition between two opposing effects (the introduction of beneficial light-harvesting units and the gradual disruption of the long-range order and crystallinity of the COF framework). Are there any results or previous studies that support this phenomenon?

Response to comment:

Thank you for this insightful comment. We have carefully re-examined the literature and found that non-monotonic activity trends arising from the interplay between functional unit incorporation and the preservation of long-range structural order are frequently reported in COF-based photocatalysis. To clarify this point, we have revised the manuscript and added relevant references. The supporting evidence is summarized below.

First, a recent Nature Communications study explicitly demonstrated that COF photocatalytic performance is highly sensitive to crystallinity and exhibits an optimal modulation window (*Nat. Commun.*, **2025**, 16, 1940). In that work, moderate structural regulation enhanced charge delocalization, prolonged carrier lifetimes, and improved interfacial charge transfer. In contrast, excessive structural perturbation reduced crystallinity and led to inferior photocatalytic activity. Notably, the authors showed that over-regulation disrupts long-range order, suppressing charge transport even when light absorption remains largely unchanged.

Second, defect-engineering studies further support this competition effect. For example, Yao et al. systematically tuned the defect density in pyrene-based COFs and observed a clear volcano-type dependence of photocatalytic performance on defect concentration (*Adv. Sci.*, **2025**, e18948). Moderate defect introduction improved visible-light absorption and charge separation, whereas excessive defects degraded structural integrity and offset the performance gains achieved at lower defect levels.

In addition, a recent study published in ACS Materials Letters provides a direct parallel to our observations (*ACS Mater. Lett.*, **2023**, 5, 2877–2886). In this work, Yang et al. introduced localized defects into an olefin-linked COF (TtBda) by incorporating monoaldehyde electron donors (e.g., pyrene). The photocatalytic activity increased with increasing defect concentration, reaching a maximum at approximately 20% loading, but declined upon further introduction of defects. The authors attributed this decrease to a gradual loss of crystallinity caused by the absence of sufficient framework connectors, which hindered the formation of long-range ordered structures.

Similar non-monotonic trends have also been reported in related COF systems (*J. Mater. Chem. A*, **2021**, 9, 25474; *ACS Appl. Polym. Mater.*, **2025**, 7, 4525–4534), highlighting the general importance of balancing functional unit incorporation with structural order.

In summary, these precedents support our interpretation that the observed non-monotonic H₂O₂ production originates from a balance between two opposing effects: (i) enhanced light harvesting and charge generation enabled by functional unit incorporation, and (ii) gradual disruption of long-range order and crystallinity that limits charge transport at higher doping levels. We have clarified this mechanism and incorporated the above references into the revised manuscript. The details can be found in the revised manuscript and the revised supplementary information, as follows:

Page 13 in the revised manuscript: “The observed trend may result from a competition between enhanced photon harvesting and a progressive amorphization of the COF scaffold induced by the incorporation of defective engineering (Supplementary Fig. 38). (*Nat. Commun.*, **2025**, 16, 1940; *Adv. Sci.*, **2025**, e18948; *ACS Mater. Lett.*, **2023**, 5, 2877–2886; *J. Mater. Chem. A*, **2021**, 9, 25474; *ACS Appl. Polym. Mater.*, **2025**, 7, 4525–4534)

Supplementary Fig. 38 Experimentally obtained PXRD patterns of COFs.

About Question (2),

The authors should add an explanation of structural stability after 120 hours in the revised manuscript, in the same manner as they did after 96 hours.

Response to comment:

We thank the reviewer for this valuable suggestion. Following the same analytical framework used for the 96 h stability evaluation, we have now added a detailed discussion of the structural stability after 120 h of photocatalytic reaction in the revised manuscript. These results show that although COF-300-D-F has excellent long-term structural and performance stability (96 h), its long-range ordered structure is destroyed after 120 h of photocatalytic reaction. This explanation has been included in the revised manuscript to provide a consistent and comprehensive evaluation of structural stability in different reaction durations. The details can be found in the revised manuscript and the revised supplementary information, as follows:

Page 15 in the revised manuscript: “...However, after 120 h of photocatalytic reaction, the long-range ordered structure of COF-300-D-F underwent obvious degradation. This was evidenced by the disappearance of sharp peaks in the PXRD pattern and the morphological deterioration of the particles observed by SEM. In contrast, the IR spectrum showed that C=N linkages (1624 cm^{-1}) were still retained, collectively suggesting that the crystalline framework was largely collapsed into short-range domains after photocatalysis for 120 h.”

Supplementary Fig. 81 The long-term photocatalytic reaction of COF-300-D-F to produce H₂O₂ at different reaction times was characterized by PXRD (a), FTIR (b), and SEM (c).

About Question (3),

Although the authors carefully improved their experimental method, Figure 4g still shows an increase in H₂O₂ production per unit time after 24 hours; therefore, the authors should explain this phenomenon and its underlying mechanism.

Response to comment:

We thank the reviewer for this insightful comment and fully agree that the increase in the apparent H₂O₂ production rate after 24 h in Fig. 4g deserves careful discussion. We provide the following mechanistic explanation.

First, unlike many reported 2D COFs that rely on π - π stacking and often suffer from partial delamination, pore collapse, or active-site deactivation during prolonged photocatalysis, COF-300-D-F is based on a three-dimensionally interpenetrated network. This architecture endows the framework with markedly enhanced structural robustness under long-term reaction conditions. Consistent with this, no detectable structural degradation is observed over the entire 96 h photocatalytic period, as evidenced by the PXRD reflections, FTIR, and SEM images after reaction (Supplementary Fig. 81). As a result, the catalytic activity does not exhibit the gradual decay commonly observed for less stable 2D COF systems at extended reaction times (*Nat. Commun.*, **2024**, 15, 8023). Instead, the maintained structural integrity allows the system to sustain—and even slightly improve—its effective reaction rate in the later stage.

Second, we note that the rate enhancement observed after 24 h is pronounced. Given the intrinsically complex and tortuous pore architecture of 3D COFs, we attribute this behavior to progressive pore wetting and enhanced mass transport. The photographs taken at different time points revealed a continuous enhancement in the dispersion state of the COF particles as the reaction proceeded (Supplementary Fig. 80). At the early stage of photocatalysis, H₂O₂ generation is dominated by reactions occurring at the external surface and near-surface pores. With prolonged irradiation and continuous contact with the reaction medium, gradual wetting of the internal pore network occurs, leading to improved mass transport, enhanced diffusion of reactants, and increased accessibility of previously less-exposed active sites within the framework (*Nat. Commun.*, **2024**, 15, 8023; *Nat.*

Commun., **2025**, 16, 7654). This time-dependent increase in the effective number of participating catalytic sites reasonably accounts for the higher apparent H₂O₂ production rate at later times.

Third, we note that prolonged stirring during continuous photocatalytic operation can induce a weak but persistent emulsification-like state in the reaction system (Supplementary Fig. 80), thereby effectively increasing the solid–liquid–gas interfacial area. This effect is conceptually analogous to the Pickering emulsion catalysis mechanism, in which solid particles stabilize dynamic interfaces, thereby promoting interfacial reactions by shortening mass-transfer distances and increasing the residence time of reactants at catalytically active interfaces. The as-generated emulsion system provides the reaction interfacial area which is more than 10³-higher than the classical biphasic system, so as to accelerate various interface reactions (*Nat. Commun.*, **2025**, 16, 2490; *Angew. Chem. Int. Ed.*, **2022**, 61, e202115885; *Angew. Chem. Int. Ed.*, **2025**, 64, e202421341). Therefore, the enhancement of interfacial catalytic efficiency under long-term stirring provides an additional contribution to the increased apparent H₂O₂ production rate observed after 24 h.

Taken together, the combination of exceptional structural stability arising from the 3D interpenetrated COF framework, progressive pore wetting and enhanced mass transport, and improved interfacial mass transfer under long-term operation provides a consistent and reasonable explanation for the increase in the apparent H₂O₂ production rate after 24 h. We believe this behavior highlights a key advantage of 3D COF photocatalysts over conventional 2D COFs for sustained and long-duration photocatalytic applications. We have revised the manuscript to explicitly clarify this point and now note that the slight increase in apparent H₂O₂ production rate reflects the gradual attainment of an optimal catalytic steady state rather than catalyst degradation. We believe this clarification addresses the reviewer’s concern and improves the mechanistic transparency of the long-term photocatalytic data.

Page 14 in the revised manuscript: “...It is important to note that a slight increase in the apparent H₂O₂ production rate is observed after 24 h of continuous photocatalysis (Fig. 4g). This behavior can be attributed to the progressive pore wetting and enhanced mass transport within the 3D COF framework, together with sustained stirring that promotes a dynamic emulsified interfacial state, thereby facilitating interfacial oxygen reduction and H₂O₂ formation (Supplementary Fig. 80) (*Nat.*

Commun., **2024**, *15*, 8023; *Nat. Commun.*, **2025**, *16*, 7654; *Angew. Chem. Int. Ed.*, **2022**, *61*, e202115885; *Nat. Commun.*, **2025**, *16*, 2490).”

Fig. 4 H₂O₂ evolution performance of functional 3D COFs. **g**, Continuous and stable production of H₂O₂ over 96 hours of COF-300-D-F.

Supplementary Fig. 80 Photograph of the solution system in the long-term photocatalytic production of H₂O₂ (each image collected after stopping stirring for 10 min).

About Question (4),

Thank you for agreeing that the use of sacrificial agents diminishes the practicality of this research.

Response to comment:

We thank the reviewer's positive comment.

About Question (6),

Although the specific parameters of Pawley refinements were added to the supplementary information, the standard deviation of the lattice constants obtained by Pawley refinements should also be included.

Response to comment:

We thank the reviewer for this constructive suggestion. In response, the standard deviations of the lattice constants obtained from the Pawley refinements have now been included in the supplementary information. These values were extracted directly from the Pawley fitting procedure and provide a quantitative assessment of the refinement uncertainty. The details are as follows:

Supplementary Table 1. Model structures for COF-300, COF-300-D-15%, and COF-300-D-F were generated in BIOVIA Materials Studio 8.0. Simulation of PXRD patterns and Pawley Refinement were performed using the Reflex module.

	Space group	a	c	R_{wp}	R_p
COF-300	$I4_1/a$	27.23544 ± 0.00055	7.51159 ± 0.00031	2.68%	1.84%
COF-300-5%	$I4_1/a$	27.14229 ± 0.00063	7.54181 ± 0.00037	1.75%	1.26%
COF-300-10%	$I4_1/a$	27.15163 ± 0.00066	7.54749 ± 0.00039	1.59%	1.15%
COF-300-15%	$I4_1/a$	27.11790 ± 0.00067	7.55957 ± 0.00038	1.68%	1.24%
COF-300-20%	$I4_1/a$	27.10013 ± 0.00107	7.55770 ± 0.00058	1.39%	1.01%
COF-300-D-F	$I4_1/a$	27.07696 ± 0.00086	7.56828 ± 0.00047	1.54%	1.13%

We believe these supplementary details ensure clarity and repeatability in the in-depth revision process. Thank you again for helping us improve our paper.

Reviewer #2 (Remarks to the Author):

“The authors have provided a thorough and convincing response to all my comments. I am satisfied with the changes and consider the paper suitable for publication in Nature Communications in its current form.”

Response to the general comment:

We are grateful to the reviewer for the time and effort devoted to evaluating our manuscript. We appreciate the reviewer’s positive feedback and are pleased that the revisions have satisfactorily addressed all concerns.

Reviewer #3 (Remarks to the Author):

“The authors have done a great job in addressing my previous concerns. Thus, I highly recommend it for publication in Nature Communications as it is.”

Response to the general comment:

We greatly appreciate the reviewer’s encouraging comments and recommendations. We are glad that the revised manuscript has satisfactorily resolved all previous concerns.

We thank all referees again for their helpful comments and suggestions, and hope that this significantly revised manuscript is now acceptable for publication in *Nature Communications*.

Best Regards,

Yours Sincerely,

Prof. Dr. Chong Cheng (on behalf of the authors)